# DISENTANGLING PRIMITIVE REPRESENTATION STRUCTURES FOR IMAGE GENERATION

## ABSTRACT

This paper explains a neural network for image generation from a new perspective, *i.e.*, explaining representation structures for image generation. We propose a set of desirable properties to define the representation structure of a neural network for image generation, including feature completeness, spatial boundedness and consistency. These properties enable us to propose a method for disentangling primitive feature components from the intermediate-layer features, where each feature component generates a primitive regional pattern covering multiple image patches. In this way, the generation of the entire image can be explained as a superposition of these feature components. We prove that these feature components, which satisfy the feature completeness property and the linear additivity property (derived from the feature completeness, spatial boundedness, and consistency properties), can be computed as OR Harsanyi interaction. Experiments have verified the faithfulness of the disentangled primitive regional patterns.

## 1 INTRODUCTION

While we have acknowledged the extensive research on designing neural networks with intrinsically interpretable features for image generation (Li et al., 2022; Xing et al., 2020), in this paper, we limit our discussion to the post-hoc explanation of trained deep neural networks (DNNs). To this end, most studies have predominantly focused on controlling image generation through modifications of input codes (Härkönen et al., 2020; Voynov & Babenko, 2020) or intermediate features (Bau et al., 2018; Yüksel et al., 2021; Shi et al., 2025; Li et al., 2024a).

In contrast to superficially analyzing the meaning of a single feature vector in isolation, we aim for a more granular and nuanced explanation[1], and focus on **a new problem, *i.e., can we extract a concise and interpretable representation structure, which encompasses all feature components, to precisely explain how a neural network generates an image?*** To this end, how to rigorously define the representation structure for image generation and whether a neural network does have such an interpretable representation structure remain open questions.

In this paper, we aim to disentangle the features in an intermediate layer of a DNN into different primitive feature components: $\Delta \boldsymbol{f}_1 + \Delta \boldsymbol{f}_2 + \cdots + \Delta \boldsymbol{f}_m$. As Figure 1 shows, each feature component $\Delta \boldsymbol{f}_k$ is responsible for generating a regional pattern (within image patches in $A_k$). $A_k$ is termed the *action field* of $\Delta \boldsymbol{f}_k$, i.e., the specific image region influenced by $\Delta \boldsymbol{f}_k$. Accordingly, the generation of the entire image can be explained as the superposition of all the primitive feature components. Compared to mechanistic interpretability studies that explore visual patterns across numerous neurons (Härkönen et al., 2020; Yüksel et al., 2021; Bau et al., 2018), our method rigorously decomposes the responsibility for generating each patch into distinct feature components, without expecting individual neurons to autonomously represent semantically clear concepts.

To achieve the above objective, we first establish a set of properties that a faithful explanation of representation structures must satisfy, and characterize these essential properties as three axioms. **(1)** The feature completeness property means that the intermediate feature $\boldsymbol{f}$ is supposed to be precisely expressed as the sum of all its feature components. **(2)** The spatial boundedness property means that each feature component $\Delta \boldsymbol{f}_k$ should only influence patches within its action field $A_k$. **(3)** The consistency property means that each feature component $\Delta \boldsymbol{f}_k$ should consistently generate the

---

[1]Appendix A introduces a more detailed comparison between our method and related work.

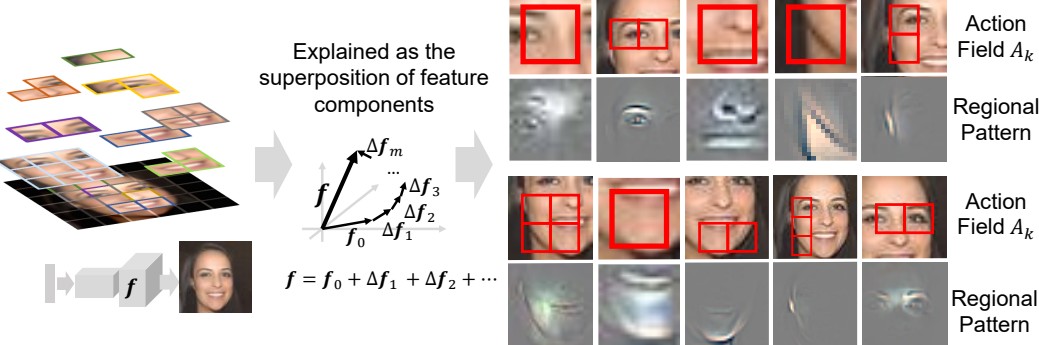

Figure 1: Feature components disentangled from a DNN for image generation, which can be considered an explanation of the DNN's internal representation structure. Each feature component $\Delta \boldsymbol{f}_k$ generates a primitive regional pattern covering a specific set $A_k$ of patches in the image $\mathbf{x}$. $A_k$ is termed the *action field* of the feature component. The generation of the image $\mathbf{x}$ can therefore be mathematically represented as the superposition of these regional patterns.

same regional pattern $\Delta \mathbf{x}_k$, regardless of the generation of its neighboring regions. We further derive the linear additivity property based on the above three properties, which further clarifies that the generation of a specific region could be explained as the linear superposition of relevant feature components. Representation structures failing to meet these properties usually cause contradictions that are mathematically and cognitively significant.

Building upon these axiomatic properties, we propose a method to disentangle each feature component as an OR interaction[2] between demands of reconstructing a set of patches. We mathematically prove that feature components disentangled by our method automatically satisfy both the feature completeness property and the linear additivity property. We further conducted experiments to verify that the extracted feature components fulfilled the proposed spatial boundedness property and consistency property.

In addition, we observed three main advantages of the feature components. **(1) Sparsity**: the extracted feature components exhibited high sparsity, *i.e.*, only a few elements in feature components contributed to image generation, while the remaining elements typically exerted negligible influence. **(2) Transferability**: the representation structure demonstrated transferability across different images generated by the same network, *i.e.*, the action fields of salient feature components were transferable across samples. **(3) Controllable reconstruction**: the extracted feature components enabled control over the sequential reconstruction of specific regions.

The contributions of this paper are summarized as follows. **(1)** We define three axiomatic properties that a faithful representation structure for image generation is supposed to satisfy. **(2)** Based on these properties, we derive a method to disentangle each feature component as an OR interaction between the demands of reconstructing different patches, and we prove that the feature component satisfies the feature completeness and linear additivity properties. **(3)** Experimental results demonstrated that the feature components satisfied the spatial boundedness and consistency properties. **(4)** The interactions' high sparsity, transferability, and the capacity for controlling reconstruction were also verified in experiments.

## 2 ALGORITHM

### 2.1 AXIOMATIC PROPERTIES FOR REPRESENTATION STRUCTURE

In recent years, although many studies (Härkönen et al., 2020; Voynov & Babenko, 2020; Yüksel et al., 2021; Shi et al., 2025; Li et al., 2024a; Bau et al., 2018) have explored the internal logic learned by image-generating DNNs, most approaches remain superficial. For example, most studies (Härkönen et al., 2020; Voynov & Babenko, 2020; Yüksel et al., 2021; Bau et al., 2018) identify

---

[2]The OR interaction means that if any patch in $A_k$ needs to be reconstructed, this feature component $\Delta \boldsymbol{f}_k$ must be added to the intermediate-layer feature.

| Property | GANSpace (2020) | GAN Dissection (2018) | GradCtrl (2022) | Interpret Diffusion (2024b) | Asyrp (2022) | Ours |
|---|---|---|---|---|---|---|
| Feature Completeness | | ✓ | | | ✓ | ✓ |
| Spatial Boundedness | | | ✓ | | | ✓ |
| Consistency | | | | | ✓ | ✓ |

Table 1: Comparison of explanation methods across the three axiomatic properties (Section 2.1). See Appendix C for analysis.

specific feature vectors to control the generated image. However, the patterns modeled by neural networks inherently exhibit hierarchical representation structures that are far more complex than feature vectors. Therefore, we propose three properties in three axioms that any faithful explanation of an image-generating DNN is supposed to satisfy, thereby ensuring the mathematical rigor of the explanation. Although previous explanation studies were not designed based on the three properties, we can still compare them in terms of our axiomatic properties. Please see Table 1 for details.

Before the detailed introduction of the three axiomatic properties, let us first provide an overview of the problem setting for the representation structure. When we input a certain code $z$ to generate an image $\mathbf{x}$, we can rewrite the generated image $\mathbf{x}$ as an aggregation of $n$ image patches, *i.e.*, $\mathbf{x} = [\mathbf{x}_1, \ldots, \mathbf{x}_n]^T$, where $\mathbf{x}_i$ represents the $i$-th patch. We use $N = \{1, 2, \ldots, n\}$ to denote the index set of all image patches. Let $\boldsymbol{f} \in \mathbb{R}^D$ denote the feature in an intermediate layer of the DNN computed given the input $z$, *i.e.*, $z \to \boldsymbol{f} \to \mathbf{x}$.

This allows us to reformulate the image generation process as a reconstruction from the intermediate feature $\boldsymbol{f}$. The goal of the explanation is illustrated in Axiom 1, which decomposes the feature $\boldsymbol{f}$ into a set of feature components: $\boldsymbol{f} \xrightarrow{\text{decompose}} \Delta\boldsymbol{f}_1 + \Delta\boldsymbol{f}_2 + \cdots + \Delta\boldsymbol{f}_m$, where each feature component $\Delta\boldsymbol{f}_k$ is used to reconstruct a specific regional pattern $\Delta\mathbf{x}_k$.

In this way, we propose the following three axioms to ensure the faithfulness of feature decomposition. The first axiom presents the *feature completeness* property. We require the representation structure to account for all feature components in the network, rather than selectively identifying only a few meaningful feature vectors that control the image output while ignoring other features (Härkönen et al., 2020; Voynov & Babenko, 2020; Yüksel et al., 2021). Thus, Axiom 1 (the feature completeness property) requires that the linear superposition of all feature components $\Delta\boldsymbol{f}_k$ should equal the intermediate feature $\boldsymbol{f}$ needed for generating image $\mathbf{x}$.

**Axiom 1.** *(Feature completeness property)*

$$\boldsymbol{f} = \boldsymbol{f}_0 + \sum_{k \in M} \Delta\boldsymbol{f}_k, \tag{1}$$

*where $M = \{1, 2, \ldots, m\}$ denotes the index set of all feature components. $\boldsymbol{f}_0$ represents a baseline feature encoding "no information."* [3]

The second axiom presents the *spatial boundedness* property. Each feature component $\Delta\boldsymbol{f}_k$ is supposed to exclusively generate a specific set of image patches, namely the *action field*, denoted by $A_k$. All other patches in $N \setminus A_k$ should not be affected by $\Delta\boldsymbol{f}_k$.

**Axiom 2.** *(Spatial boundedness property)*

$$\forall k \in M, \; \forall \Omega \subseteq M \setminus \{k\}, \; \text{region}(N \setminus A_k \mid \mathcal{G}(\Omega)) = \text{region}(N \setminus A_k \mid \mathcal{G}(\Omega \cup \{k\}))$$

$$\text{s.t.} \quad \mathcal{G}(\Omega) \stackrel{def}{=} g(\boldsymbol{f}_0 + \sum_{l \in \Omega} \Delta\boldsymbol{f}_l), \tag{2}$$

*where $g(\cdot)$ denotes the function that maps the intermediate feature to the output image, $\mathcal{G}(\Omega)$ denotes the image generated by adding all feature components in $\Omega$ to the intermediate layer, and $\text{region}(N \setminus A_k \mid \mathcal{G}(\Omega))$ denotes an operation function that clips image patches in $N \setminus A_k$ from the generated image. $\Omega$ denotes a set of feature components.*

---

[3] $\boldsymbol{f}_0$ can be set as the average feature over all features given different input codes $z$. i.e., $\boldsymbol{f}_0 = \mathbb{E}_z[d(z)]$, where $d(\cdot)$ denotes the modules that uses the input code $z$ to generate the intermediate feature.

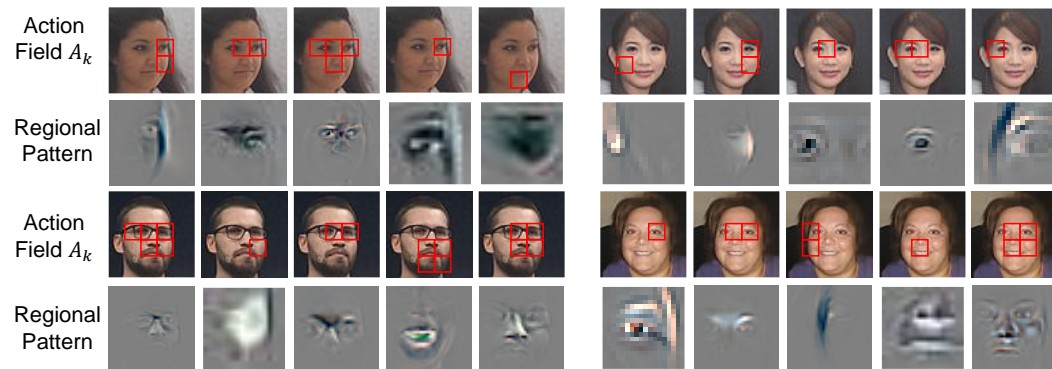

Figure 2: Visualization of regional patterns $\Delta\mathbf{x}_k$ corresponding to different feature components $\Delta\boldsymbol{f}_k$ extracted from the SiD Diffusion Model. Please see Appendix G for more results on other image generation networks.

The third axiom presents the *consistency* property. In a faithful feature decomposition, each feature component $\Delta\boldsymbol{f}_k$ is supposed to consistently add the same regional pattern $\Delta\mathbf{x}_k$ to the generated image, no matter how other feature components are set.

**Axiom 3.** *(Consistency property)*

$$\forall k \in M, \quad \forall\Omega \subseteq M \setminus \{k\}, \quad \mathcal{G}(\Omega \cup \{k\}) - \mathcal{G}(\Omega) = \Delta\mathbf{x}_k. \tag{3}$$

Therefore, according to Axioms 1, 2 and 3, we can further derive a theorem: each subset of image patches in $S \subseteq N$ can well be generated by adding all feature components whose action fields cover (or partially cover) $S$. This is termed the *linear additivity* property of image generation.

**Theorem 1-$\alpha$.** *(Linear additivity property, derived in Appendix D)*

$$\forall S \subseteq N, \ \text{region}(S \mid g(\boldsymbol{f}')) = \text{region}(S \mid \mathbf{x}) \quad s.t. \quad \boldsymbol{f}' = \boldsymbol{f}_0 + \sum\nolimits_{k \in M: A_k \cap S \neq \emptyset} \Delta\boldsymbol{f}_k, \tag{4}$$

where $\text{region}(S \mid \mathbf{x})$ *denotes an operation that clips image patches in S from the image* $\mathbf{x}$.

In this way, the aforementioned axiomatic properties guarantee a clear internal representation structure for image generation, and the entire image generation can be decomposed into the superposition of different regional patterns. *I.e.*, $\mathbf{x} = g(\boldsymbol{f}_0 + \sum_{k \in M} \Delta\boldsymbol{f}_k)$. In contrast, existing interpretability methods (see Table 1) that manipulate features for image edits do not account for the network's detailed internal structure.

## 2.2 EXTRACTION OF THE REPRESENTATION STRUCTURE

**Minimal Feature**. In this subsection, we will further develop a new method to disentangle feature components. Before presenting our method, we define the minimal feature $\hat{\boldsymbol{f}}_S$. It provides the least information required to generate all patches in $S \subseteq N$. It can be computed as the minimal feature variation from the baseline state $\boldsymbol{f}_0$ that still allows full reconstruction of $S$:

$$\hat{\boldsymbol{f}}_S \overset{\text{def}}{=} \text{argmin}_{\boldsymbol{f}_S} \|\boldsymbol{f}_S - \boldsymbol{f}_0\|_1 \quad \text{s.t.} \quad \tilde{\mathbf{x}} = g(\boldsymbol{f}_S), \ \| \text{region}(S \mid \tilde{\mathbf{x}}) - \text{region}(S \mid \mathbf{x})\|_2 < \epsilon, \tag{5}$$

where the constraint $\| \text{region}(S \mid \tilde{\mathbf{x}}) - \text{region}(S \mid \mathbf{x})\|_2 < \epsilon$ ensures the reconstruction quality. $\epsilon$ is a small positive threshold. This formulation leads to a classical optimization problem that can be efficiently solved. Please refer to Appendix E for further details regarding the value of $\epsilon$ and the optimization procedure.

**Computing feature components as OR interactions.** According to Theorem 1-$\alpha$, each feature component $\Delta\boldsymbol{f}_k$ can be formulated as an OR interaction among the reconstruction demands of different image patches within its action field $A_k$. To make this OR interaction more explicit, we reformulate Theorem 1-$\alpha$ as Theorem 1-$\beta$.

**Theorem 1-$\beta$.** *(OR relationship, proof in Appendix D) Given a demand of generating a set of image patches $S \subseteq N$, the feature component $\Delta\boldsymbol{f}_k$ with an action field $A_k = \{i_1, i_2, \ldots, i_l\}$ must be added to the baseline feature $\boldsymbol{f}_0$, if any patch $i_1$, or $i_2$, or, $\ldots$, or $i_l$ is contained within S.*

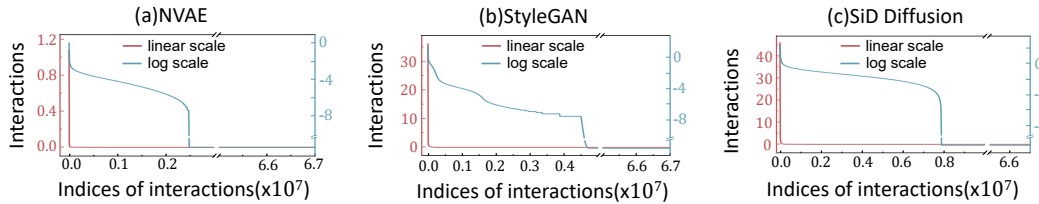

Figure 3: Sparsity of the extracted feature components. We show absolute effects of all interaction elements in all feature components, sorted in descending order. Both the original and log scale curves are shown.

Therefore, the aforementioned feature components $\{\Delta \boldsymbol{f}_k\}$ can be formulated as the Harsanyi OR interaction (Zhou et al., 2023) among patches in $S$. In particular, we consider the DNN uses the minimal features $\{\hat{\boldsymbol{f}}_S\}$ in Equation (5) to generate patches in $A_k$. Then, the Harsanyi OR interaction among generation demands of different patches in $A_k \subseteq N$ can be computed as

$$\Delta \boldsymbol{f}_k \stackrel{\text{def}}{=} \boldsymbol{I}_{or}(A_k) = -\sum\nolimits_{S' \subseteq A_k} (-1)^{|A_k| - |S'|} \cdot \hat{\boldsymbol{f}}_{N \setminus S'}, \tag{6}$$

where $\hat{\boldsymbol{f}}_{N \setminus S'}$ is given as Equation (5). The optimization of $\hat{\boldsymbol{f}}_{N \setminus S'}$ is introduced in Appendix E.

**Feature components (OR interactions) satisfy the feature completeness property (in Axiom 1) and the linear additivity property (in Theorem 1-$\alpha$, derived from Axioms 1-3).** In terms of the linear additivity property in Theorem 1-$\alpha$, we prove Theorem 2, which shows that the extracted feature components in Equation (6) can be reformulated to satisfy the linear additivity property. Specifically, we construct a logical model $h(S)$ that superimposes all feature components whose action fields cover (or partially cover) patches in the patch set $S$:

$$h(S) = \boldsymbol{f}_0 + \sum\nolimits_{k \in M} \mathbb{1}(i_1 \in A_k \text{ or } i_2 \in A_k \dots \text{ or } i_l \in A_k) \cdot \Delta \boldsymbol{f}_k, \tag{7}$$

where the binary trigger function $\mathbb{1}(\cdot)$ returns 1 if any image patch in the action field $A_k$ is covered by the set $S$ (as required in Theorem 1-$\beta$). Otherwise, it returns 0.

Theorem 2 demonstrates that the logical model $h(S)$ well matches the minimal feature $\hat{\boldsymbol{f}}_S$ for reconstructing patches in $S$.

**Theorem 2.** *(proof in Appendix D) Given a random set of image patches $S \subseteq N$, the logical model $h(S)$ superimposes all triggered feature components to obtain the minimal feature of reconstructing patches in $S$. I.e., $\forall S \subseteq N, \hat{\boldsymbol{f}}_S = h(S)$ and $\| \operatorname{region}(S \mid \tilde{\mathbf{x}}) - \operatorname{region}(S \mid \mathbf{x}) \|_2 < \epsilon$, subject to $\tilde{\mathbf{x}} = g(h(S))$.*

For the examination of the feature completeness property in Axiom 1, if we set $S = N$ in Theorems 2, it is equivalent to the feature completeness property in Axiom 1. It proves that the linear superposition of all extracted feature components $\Delta \boldsymbol{f}_k$ equals the intermediate feature $\boldsymbol{f}$ needed for generating image $\mathbf{x}$. We will demonstrate in Sections 3.2 and 3.3 that the extracted feature components $\Delta \boldsymbol{f}_k$ also satisfy the spatial boundedness property presented in Axiom 2 and the consistency property presented in Axiom 3.

**Visualizing the regional pattern of each feature component.** The regional pattern is calculated as the average change of the image generation result when the feature component $\Delta \boldsymbol{f}_k$ is removed from the feature. This corresponds to extracting the image region within its associated action field $A_k$ from the difference image $\Delta \mathbf{x} = E_{\Omega \subseteq N \setminus \{k\}}[\mathcal{G}(\Omega \cup \{k\}) - \mathcal{G}(\Omega)]$. Figure 2 visualizes regional patterns corresponding to the top-10% most salient feature components (those with the highest L2-norms $\|\Delta \boldsymbol{f}_k\|_2$). We observe that the regional pattern generated by a feature component $\Delta \boldsymbol{f}_k$ is typically located within its action field $A_k$.

### 2.3 DOES A DNN HAVE A CLEAR REPRESENTATION STRUCTURE?

Although the feature components $\Delta \boldsymbol{f}_k$ are proven to satisfy the feature completeness property in Axiom 1 and the linear additivity property in Theorem 1-$\alpha$ (derived from the spatial boundedness

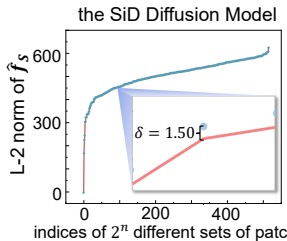 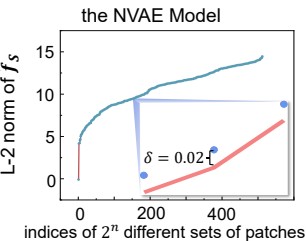

Figure 4: Verifying the linear additivity property. The L2-norm of different minimal features is shown in ascending order. The dots above represent the matching error between the actual minimal feature and the feature estimated by the logical model $h(S)$. The matching error was just about 0.3%-0.5% of the L2-norm.

Axiom 2 and the consistency Axiom 3), we still need to verify whether all DNNs have clear representation structures for image generation. To this end, we examine the sparsity of extracted feature components, which is crucial to a clear representation structure. Dense feature components are usually not considered as a faithful explanation[4]. We follow Ren et al. (2024a), Li & Zhang (2023) and Chen et al. (2024) to define the sparsity of interactions as a condition where most elements of all the feature components (*i.e.,* all interactions $\{\Delta \boldsymbol{f}_k = I_{or}(A_k)\}$) have negligible values, while only a few elements of feature components have salient values[5].

Figure 3 visualizes all elements of the extracted interactions (*i.e.,* feature components) by sorting their absolute effects in descending order. Most elements of feature components are almost zero, which verifies the sparsity of the extracted feature components. This result shows that a well-trained DNN uses only sparse feature components for image generation.

## 3 EXPERIMENT

In this section, we conducted experiments on four deep neural networks for image generation to extract their representation structures, including (1) a one-step diffusion model based on Score Identity Distillation (SiD) (Zhou et al., 2024b) trained on the FFHQ dataset (Karras et al., 2019) at $64 \times 64$ resolution, (2) the NVAE model (Vahdat & Kautz, 2020) trained on the CelebA $64 \times 64$ dataset (Liu et al., 2015), (3) the BigGAN model (Brock, 2018) trained on ImageNet (Deng et al., 2009) at $128 \times 128$ resolution, and (4) the StyleGAN model (Karras et al., 2020) trained on the AFHQ Cat dataset (Choi et al., 2020) at $512 \times 512$ resolution. Specifically, given an input code $\boldsymbol{z}$, each DNN generates an image $\mathbf{x}$, with an intermediate feature $\boldsymbol{f}$ at a certain layer. We analyzed $\boldsymbol{f} = d(\boldsymbol{z})$ in the following layers: *layer44* and *layer66* in the decoder of NVAE; *layer6* and *layer8* in BigGAN-128; *layer3* and *layer6* in the synthesis network of StyleGAN2-512; and *layer32x32up* in the decoder of the SiD diffusion model. We extended the software package released by Zhou et al. (2023) to extract OR interactions. The baseline feature component $\boldsymbol{f}_0$ was computed by averaging features over all input codes: $\boldsymbol{f}_0 = \mathbb{E}_{\boldsymbol{z}}[d(\boldsymbol{z})]$. We segmented the generated image $\mathbf{x}$ into $6 \times 6$ patches and followed Li & Zhang (2023) to randomly select $n = 9$ patches on the foreground object to reduce computational cost, as interaction computation was NP-complete. Finally, we computed OR interactions (*i.e.,* feature components) required for reconstructing these patches.

### 3.1 VERIFYING THE LINEAR ADDITIVITY PROPERTY

In Section 2, we have proven that the feature components extracted using OR interactions defined in Equation (6) satisfy the linear additivity property stated in Theorem 1-$\alpha$. Nevertheless, we conducted experiments to validate the linear additivity property.

Specifically, we examined whether the extracted feature components could reconstruct each minimal feature $\hat{\boldsymbol{f}}_S$. If the reconstructed minimal feature $h(S)$ accurately matched the real minimal feature $\hat{\boldsymbol{f}}_S$, then we considered the linear additivity property successfully validated. Thus, we used $\delta = \|h(S) - \hat{\boldsymbol{f}}_S\|_1 / \dim(\hat{\boldsymbol{f}}_S)$ to measure the matching error, where $\dim(\hat{\boldsymbol{f}}_S)$ denotes the dimensionality.

Figure 4 shows the L2-norm of all minimal features $\hat{\boldsymbol{f}}_S$, sorted in ascending order. The matching error $\delta$ is plotted as dots above the curve. To simplify the calculation, we selected the top-30 feature

---

[4]As discussed in Appendix I, dense feature components usually correspond to chaotic noises in features.

[5]In the field of the interaction theory, the sparsity is defined as the state that the vast majority of interaction values are negligible, with sparse interaction values being significant. Please see Appendix H.

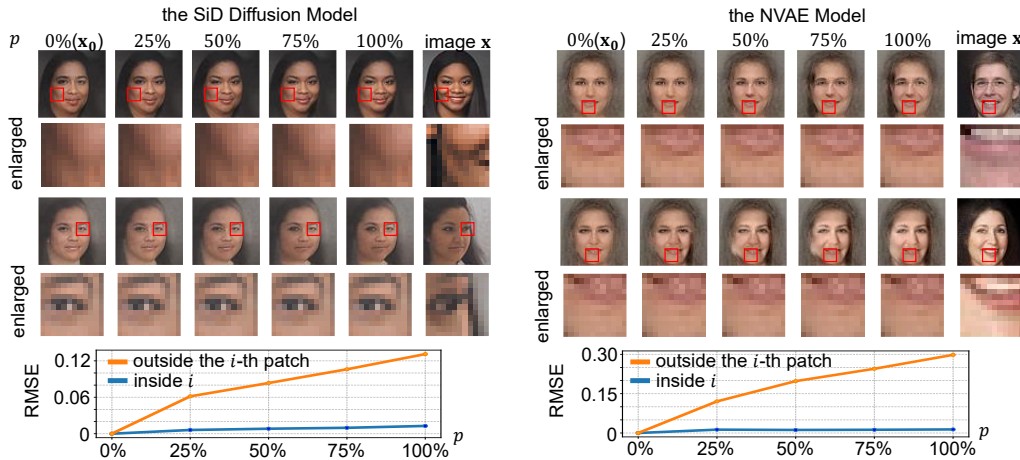

Figure 5: Verifying the spatial boundedness property by checking boundary breach from neighboring feature components. We added different ratios $p$ of feature components whose action fields did not cover the selected image patch $i$ (red boxes). We computed the reconstruction error (i.e. RMSE) between the $i$-th patch in the generated image $\tilde{\mathbf{x}}$ and the same patch in the base image $\mathbf{x}_0$. The RMSE of other patches in $N \setminus \{i\}$ between $\tilde{\mathbf{x}}$ and $\mathbf{x}_0$ was also reported. Despite the reconstruction effects on other regions, the selected patch $i$ was not reconstructed. Appendix G shows more results.

components with the highest L2-norm to match the minimal features. Please see Appendix E for the detailed settings. The tiny matching error verified the linear additivity property.

## 3.2 VERIFYING THE SPATIAL BOUNDEDNESS PROPERTY

While Theorem 1-$\alpha$ and the experiments in Section 3.1 confirm the linear additivity property, we now test the spatial boundedness and consistency properties, which are more fundamental and used to derive Theorem 1-$\alpha$. We first examine the spatial boundedness property in Axiom 2, *i.e.,* whether feature components $\Delta \boldsymbol{f}_k$ exclusively generate patches within action fields $A_k$.

### 3.2.1 EXAMINING THE REAL ACTION FIELD OF A FEATURE COMPONENT

In the first experiment, we visualized the regional pattern $\Delta \mathbf{x}_k$ generated by each feature component $\Delta \boldsymbol{f}_k$, so as to check whether the regional pattern $\Delta \mathbf{x}_k$ only influenced patches within its action field $A_k$. The pixel-wise differences of the pattern $\Delta \mathbf{x}_k$ were termed the real action field. This property was evidenced by showing that each regional pattern $\Delta \mathbf{x}_k$ only influenced image patches within the expected action field $A_k$, without affecting patches outside $A_k$. The detailed experimental settings are outlined in Section 3. The relative strength of the regional pattern $\Delta \mathbf{x}_k$ within the action field $A_k$ to the significance of the entire pattern $\Delta \mathbf{x}_k$ was quantified as $\gamma = \| \operatorname{region}(A_k \mid \Delta \mathbf{x}_k) \|_2 / \| \Delta \mathbf{x}_k \|_2$.

Figure 6 visualizes regional patterns and shows their corresponding $\gamma$ values. The results showed that the image region affected by each regional pattern $\Delta \mathbf{x}_k$ was predominantly contained within its action field $A_k$ (all the $\gamma$ values for all regional patterns were greater than 97.9%). This verified that the extracted feature components satisfied the spatial boundedness property.

### 3.2.2 CHECKING BOUNDARY BREACH FROM NEIGHBORING FEATURE COMPONENTS

To verify the spatial boundedness property in Axiom 2, we conducted experiments to check whether the generation of an image patch would be mistakenly influenced by its neighboring feature components, i.e., those with action fields near but not covering the patch. Given a random patch $i \in N$, and we obtained the reconstructed image based on these feature components $\tilde{\mathbf{x}} = \mathcal{G}(\Omega_{(i)})$, subject to $\Omega_{(i)} = \{k \mid A_k \not\supseteq i\}$. We calculated the RMSE $\| \operatorname{region}(\{i\}|\tilde{\mathbf{x}}) - \operatorname{region}(\{i\}|\mathbf{x}_0) \|_2$ to measure the influence of neighboring feature components on the $i$-th image patch *w.r.t.* the base image $\mathbf{x}_0$ generated by the baseline feature $\boldsymbol{f}_0$.

Figure 5 demonstrates the generation effect observed when we incrementally added feature components from $\Omega_{(i)}$. Results showed that when we gradually added feature components from $\Omega_{(i)}$, the

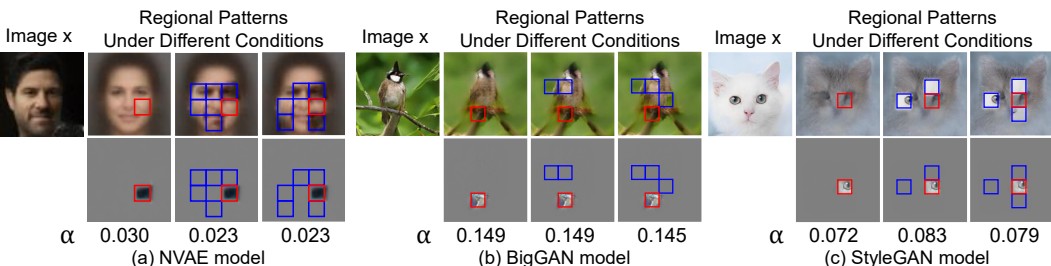

Figure 6: Verifying spatial boundedness of a regional pattern. For each image $\mathbf{x}$, we visualize the regional patterns corresponding to different feature components, with their action fields $A_k$ highlighted by red bounding boxes. The relative strength $\gamma$ within the action field is reported below each regional pattern. *Please see Appendix G for additional results on more models and samples.*

Figure 7: Verifying the consistency property of the representation structure. For each image $\mathbf{x}$, given a feature component $\Delta \boldsymbol{f}_k$ with its corresponding action field $A_k$ (in red bounding boxes), we visualize the regional pattern generated by $\Delta \boldsymbol{f}_k$, under the condition that the patches within the blue bounding boxes had already been generated. Under varying conditions, the regional patterns $\Delta \mathbf{x}_k$ generated by $\Delta \boldsymbol{f}_k$ exhibited consistent visual appearance. We also report an inconsistency score $\alpha$ for each regional pattern.

RMSE value of the $i$-th patch remained close to zero, confirming that this patch remained largely unaffected. Meanwhile, the RMSE values for patches outside the $i$-th patch increased, which reflected progressive generation of those regions. These findings confirmed that feature components, whose action fields excluded the $i$-th patch $\mathbf{x}_i$, had negligible influence on the generation of $\mathbf{x}_i$. This successfully verified the spatial boundedness property of feature components.

### 3.3 VERIFYING THE CONSISTENCY PROPERTY OF THE REPRESENTATION STRUCTURE

We conducted experiments to verify the consistency property in Axiom 3, *i.e.*, we examined whether each feature component $\Delta \boldsymbol{f}_k$ consistently added the same regional pattern $\Delta \mathbf{x}_k$ to the output image, no matter how other feature components were set. The consistency held regardless of how many patches were generated outside the action field $A_k$ of $\Delta \boldsymbol{f}_k$. We followed the detailed experimental settings in Section 3. We used $\alpha = \|\Delta \mathbf{x}_{k,S} - \Delta \mathbf{x}_{\mathrm{mean}}\|_2 / \|\Delta \mathbf{x}_{\mathrm{mean}}\|_2$ to measure the inconsistency of a regional pattern, where $S \subseteq N \setminus A_k$ denotes a set of patches outside the action field $A_k$, $\Delta \mathbf{x}_{k,S} = \mathcal{G}(\{k\} \cup \Omega_S) - \mathcal{G}(\Omega_S)$ denotes the regional pattern when patches in $S$ were generated, where $\Omega_S = \{l \mid A_l \cap S \neq \emptyset\}$ denotes the set of all feature components responsible for generating patches in $S$.

Figure 7 visualizes regional patterns generated by the same feature component $\Delta \boldsymbol{f}_k$ when different sets of other feature components in $M \setminus \{k\}$ were added. Figure 7 also shows the quantitative results of the inconsistency of regional patterns, *i.e.,* the $\alpha$ values. Both the visualization results and the small $\alpha$ values verified that the feature component $\Delta \boldsymbol{f}_k$ consistently added the same regional pattern $\Delta \mathbf{x}_k$ to the output image, which verified the consistency property of regional patterns.

### 3.4 TRANSFERABILITY OF FEATURE COMPONENTS' REGIONAL PATTERNS

After the verification of the aforementioned properties, we next test transferability, i.e., whether the extracted regional patterns remain consistent across different images of the same object. A faithful representation should reuse many of the regional patterns when generating similar images.

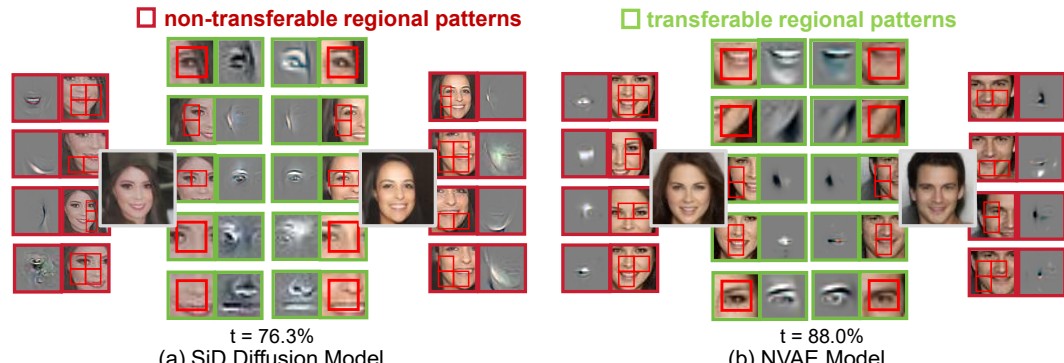

Figure 8: Visualization of transferable (green) regional patterns and non-transferable (red) regional patterns encoded by the DNN. The transferable patterns showed similar appearances on different images, and the tiny difference was caused by the tiny contextual influence. The NVAE was trained on CelebA-64 dataset, and the SiD Diffusion model was trained on FFHQ-64 dataset.

In the experiments, let us be given a pair of generated images $(\mathbf{x}^{(a)}, \mathbf{x}^{(b)})$ that contained the same object. The transferable regional patterns referred to the regional patterns saliently existed in the two images. Specifically, for image $\mathbf{x}^{(a)}$, we selected the top-$l_a$ feature components with the highest L2-norm $\|\Delta \boldsymbol{f}_k\|_2$. We used $D_{\mathbf{x}^{(a)}} = \{A_1, A_2, \ldots, A_l\}$ to denote their action fields. Similarly, given another image $\mathbf{x}^{(b)}$ containing the same object as $\mathbf{x}^{(a)}$, we also obtained another dictionary $D_{\mathbf{x}^{(b)}}$ of action fields corresponding to the top-$l_b$ feature components of $\mathbf{x}^{(b)}$. Then, we calculated $t = |D_{\mathbf{x}^{(a)}} \cap D_{\mathbf{x}^{(b)}}| / |D_{\mathbf{x}^{(a)}}|$ to quantify the proportion of regional patterns extracted from $\mathbf{x}^{(a)}$ that could be transferred to the representation of image $\mathbf{x}^{(b)}$.

Figure 8 visualizes the transferable and non-transferable regional patterns between images. The transferable patterns showed similar appearances on different images. Figure 8 also shows the proportion of transferable regional patterns, *i.e.,* the $t$ value. In the experiment, we set $l_a = 20$ and $l_b = 100$. The ratio $t$ of transferable feature components was 88.0% for the NVAE model, and 76.3% for the SiD Diffusion Model. The similarity of transferable regional patterns between images and the high $t$ values verified the transferability of feature components' regional patterns.

### 3.5 CONTROLLABLE IMAGE RECONSTRUCTION BASED ON FEATURE COMPONENTS

We conducted experiments to verify that we could use the extracted feature components to control the generation of a specific image patch. Given a randomly selected patch $i \in N$, we gradually added feature components whose action fields covered the $i$-th patch to the intermediate feature. Experiment results in Figures 9 and 10 in Appendix F showed that the target image patch $i$ was progressively reconstructed when we kept adding more feature components. Figures 11 and 12 in Appendix F also showed that we could sequentially reconstruct image patches one-by-one using feature components.

## 4 CONCLUSION

In this paper, we have proposed three axiomatic properties to explain the internal representation of image generation, which disentangles the intermediate feature into a set of feature components, each generating a specific regional pattern. We prove that the feature components satisfy the feature completeness and linear additivity properties, and can be formulated as OR interactions. In this way, the generation of the whole image can be explained as the superposition of all feature components. Experiments have validated that the extracted feature components satisfy all axioms: **(1)** Each feature component consistently generates a regional pattern that only affects patches within the action field. **(2)** The generation of a specific image patch is the superposition of regional patterns that covers the patch. **(3)** Feature components have sparse values (mostly zeros). **(4)** Regional patterns of feature components can be transferred between different generated images. **(5)** The extracted feature components enable control over the generation of specific regions.

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

## A  RELATED WORK

**The post-hoc interpretability analysis of image generation models.** In recent years, researchers have become increasingly interested in understanding how image generation models work. These models, including Generative Adversarial Networks (GANs) and diffusion models, are often considered "black boxes" because their internal decision process is difficult to understand. This has motivated many studies on interpretability, especially through post-hoc methodsthat is, techniques that analyze already-trained models without changing their architecture (Liu et al., 2016). Unlike

ante-hoc methods that build interpretability into the model during training (Li et al., 2022), post-hoc analysis offers more flexibility and is widely used. We group existing research into two main categories.

*Latent Space Manipulation and Editing*. A popular line of research focuses on controlling image generation by modifying the models latent space. Researchers find meaningful directions in this space to change attributes like age, pose, or style in the generated images. For example, Shen et al. (2020) proposed InterfaceGAN, which uses a classifier to help edit image attributes. Shen & Zhou (2021) developed SeFa, which finds interpretable directions without supervision. Other work includes StyleFlow (Abdal et al., 2021) for smooth editing, GANSpace (Härkönen et al., 2020) which uses PCA to find global directions, and StyleCLIP (Patashnik et al., 2021) that uses text to guide editing. While useful for control, these methods usually do not explain how the model internally represents information.

*Neuron- and Feature-Level Analysis*. Other studies look into individual neurons or feature maps to understand what specific parts of the model do. Bau et al. (2018) introduced GAN Dissection, which identifies neurons that detect objects like trees or doors, allowing users to add or remove those objects by changing activations. Other methods (Huang et al., 2024) use techniques like Grad-CAM (Selvaraju et al., 2017) to see which features matter during generation. These works give fine-grained insight but often miss how the entire network organizes information.

Unlike the approaches abovewhich focus on editing, our work aims to deeply understand how the entire network internally represents visual information. Previous methods are good at controlling one attribute at a time, but they don't fully explain how the model combines many elements into a coherent image. By carefully studying these internal structures, we provide new insights into how generative models work.

**Discovering internal representation structures of DNNs using symbolic explanation**. While seemingly counterintuitive, recent studies (Deng et al., 2021; Ren et al., 2021b; Li & Zhang, 2023) demonstrated that the inference patterns of DNNs could be symbolically interpreted through interaction-based frameworks. In terms of mathematical rigor, Ren et al. (2023a) established and theoretically verified that a minimal set of interactions could be used to precisely match the outputs of DNNs on any randomly masked samples. Furthermore, Ren et al. (2024a) confirmed the sparsity of interactions under three common conditions, thereby simplifying the inference logic of DNNs into a finite set of symbolic interaction patterns. Concurrently, Chen et al. (2024) proposed extracting generalizable interactions to capture shared knowledge across different models.

Building on interaction theory, researchers gradually recognized the potential of interactions to explain the representation quality of DNNs. Ren et al. (2021a) found that adversarial attacks primarily targeted complex interactions. Ren et al. (2023b) demonstrated and proved that Bayesian networks Peal (1985) faced challenges in modeling complex interactions, which likely contributed to their superior adversarial robustness. Deng et al. (2021) uncovered and illustrated that neural networks had limitations in representing complex interactions. Zhang et al. (2020) revealed that the dropout operation improved generalization by reducing the impact of interaction effects. Zhou et al. (2024a) showed that simple interactions were more likely to generalized on test samples than complex ones. Ren et al. (2024b) identified and validated the two-phase dynamics of DNNs learning interactions during training. Liu et al. (2023) provided a theoretical explanation that complex interactions posed greater learning challenges for DNNs. Utilizing interaction-based theory, Deng et al. (2024) consolidated explanations for fourteen different attribution algorithms, and Zhang et al. (2022) revealed the common mechanism behind thirteen methods that improved adversarial transferability. Shen et al. (2023) expanded the paradigm of symbolic interaction into the domain of LLMs.

In sum, most previous studies used interactions to explain neural networks with a scalar output. In comparison, in this paper, we first attempt to apply interactions to explain neural networks with a high-dimension output, i.e., the neural network for image generation, which proposes fully new challenges. To this end, we discover that each feature component disentangled from the neural network can be formulated as an OR interaction between demands of reconstructing different image regions. Experiments verified the effectiveness of the proposed method.

# B  AXIOMS AND THEOREMS FOR THE HARSANYI DIVIDEND INTERACTION

In this work, the feature components are derived through OR interaction, which constitutes a special form of the Harsanyi dividend. The Harsanyi dividend was designed as a standard metric to measure interactions between input variables encoded by the network. In this section, we present several desirable axioms and theorems that the Harsanyi dividend interaction $I(S)$ satisfies. This further demonstrates the trustworthiness of the Harsanyi dividend interaction.

The Harsanyi dividend interactions $I(S)$ satisfies the *efficiency, linearity, dummy, symmetry, anonymity, recursive* and *interaction distribution* axioms, as follows. We follow the notation in the main paper to let $u(S) = v(\mathbf{x}_S) - v(\mathbf{x}_\emptyset)$.

• **Efficiency axiom.** The output score of a model can be decomposed into interaction effects of different patterns, i.e., $u(N) = \sum_{S \subseteq N} I(S)$.

• **Linearity axiom.** If we merge output scores of two models $u_1$ and $u_2$ as the output of model $u$, i.e. $\forall S \subseteq N, u(S) = u_1(S) + u_2(S)$, then their interaction effects $I_{u_1}(S)$ and $I_{u_2}(S)$ can also be merged as $\forall S \subseteq N, I(S) = I_{u_1}(S) + I_{u_2}(S)$.

• **Dummy axiom.** If a variable $i \in N$ is a dummy variable, i.e., $\forall S \subseteq N \setminus \{i\}, u(S \cup \{i\}) = u(S)$ then it has no interaction with other variables, $\forall \emptyset \neq S \subseteq N \setminus \{i\}, I(S \cup \{i\}) = 0$.

• **Symmetry axiom.** If input variables $i, j \in N$ cooperate with other variables in the same way, i.e., $\forall S \subseteq N \setminus \{i, j\}, u(S \cup \{i\}) = u(S \cup \{j\})$, then they have same interaction effects with other variables, $\forall S \subseteq N \setminus \{i, j\}, I(S \cup \{i\}) = I(S \cup \{j\})$.

• **Anonymity axiom.** For any permutations $\pi$ on $N$, we have $\forall S \subseteq N, I_u(S) = I_{\pi u}(\pi S)$ where $\pi S = \{\pi(i) | i \in S\}$ and the new model $\pi u$ is defined by $(\pi u)(\pi S) = u(S)$. This indicates that interaction effects are not changed by permutation.

• **Recursive axiom.** The interaction effects can be computed recursively. For $i \in N$ and $S \subseteq N \setminus \{i\}$, the interaction effect of the pattern $S \cup \{i\}$ is equal to the interaction effect of $S$ with the presence of $i$ minus the interaction effect of $S$ with the absence of $i$, i.e., $\forall S \subseteq N \setminus \{i\}, I(S \cup \{i\}) = I(S | i \text{ is always present}) - I(S)$. $I(S | i \text{ is always present})$ denotes the interaction effect when the variable $i$ is always present as a constant context, i.e. $I(S | i \text{ is always present})$ $= \sum_{S \subseteq I(S)} (-1)^{|S|} \cdot u(L \cup \{i\})$.

• **Interaction distribution axiom.** This axiom characterizes how interactions are distributed for "interaction functions". An interaction function $u_T$ parameterized by a subset of variables $T$ is defined as follows. $\forall S \subseteq N$, if $T \subseteq S, u_T(S) = c$; otherwise, $u_T(S) = 0$. The function $u_T$ models pure interaction among the variables in $T$, because only if all variables in $T$ are present, the output value will be increased by $c$. The interactions encoded in the function $u_T$ satisfies $I(T) = c$, and $\forall S \neq T, I(S) = 0$.

# C  DISCUSSIONS ON WHETHER PREVIOUS WORKS SATISFIED THE THREE AXIOMATIC AXIOMS

GANSpace (Härkönen et al., 2020) operates through the application of Principal Component Analysis (PCA) to either the latent or feature space to identify significant latent directions. However, the set of image editing directions disentangled by GANSpace can only control a subset of attributes and cannot account for all the information in an image (violating the feature completeness property), while also affecting its global structure (violating the spatial boundedness property).

GAN Dissection (Bau et al., 2018) discovers a set of interpretable units strongly associated with object concepts by employing a segmentation-driven network dissection approach. The GAN Dissection method can pinpoint neurons associated with any object in an image. However, it does not guarantee that the identified neurons exclusively control the generation of the corresponding region (violating the spatial boundedness property), nor does it ensure their generation effect remains consistent regardless of the states of other neurons (violating the consistency property).

GradCtrl (Chen et al., 2022) identifies nonlinear controls through the gradient information in a learned GAN latent space, and achieves effective disentanglement coupled with multi-directional

manipulation capabilities. This method performs effectively in editing specific regions. However, its editing directions are still confined to a limited set of human-defined semantic concepts and cannot account for all the information required for image generation (violating the feature completeness property).

Interpret Diffusion (Li et al., 2024b) proposes an innovative self-supervised approach to discover interpretable latent directions for specified concepts. However, although this method can edit images by leveraging feature directions in the neural network that correspond to specific semantics, it still fails to explain the intrinsic representational structure underlying image generation (violating the feature completeness property). Moreover, the edits induced by these feature directions result in global alterations to the image (violating the spatial boundedness property).

Kwon et al. (2022) introduces Asyrp, a novel generative process that enables semantic image editing within the h-space of pretrained diffusion models. This h-space exhibits favorable propertiesincluding homogeneity, linearity, robustness, and cross-timestep consistencysimilar to those observed in the latent space of GANs. However, this method cannot ensure that the extracted feature editing directions exclusively affect the image generation of specific regions (violating the spatial boundedness property).

## D Supplementary Proofs of Theorems from the Main Paper

**Proving that the linear additivity property could be derived from Axiom 1,2,3.**

*Proof.* According to Axiom 2, each feature component $\Delta \boldsymbol{f}_k$ has its own action field $A_k$, and it is only responsible to the generation of patches in $A_k$. According to Axiom 3, each feature component $\Delta \boldsymbol{f}_k$ consistently add the same regional pattern $\Delta \mathbf{x}_k$ to the generated image, no matter how other feature components are set. Axioms 2 and 3 show that feature components are independent to each other, each $\Delta \boldsymbol{f}_k$ corresponds to a regional pattern $\Delta \mathbf{x}_k$.

According to Axiom 1, the linear superposition of all feature components $\{\Delta \boldsymbol{f}_k\}$ equals the intermediate feature $\boldsymbol{f}$ for the generation of image $\mathbf{x}$. Therefore, we can derive that the linear superposition of all regional patterns equals the image $\mathbf{x}$:

$$\mathbf{x} = \mathbf{x}_0 + \sum_{k \in M} \Delta \mathbf{x}_k \quad s.t. \quad \mathbf{x}_0 = g(\boldsymbol{f}_0) \tag{8}$$

Therefore, we can select a set of feature components $\Omega_S$ whose action fields $\{A_k\}$ cover (or partially cover) a selected image patch set $S$, i.e., $\Omega_S = \{k \mid A_k \cap S \neq \emptyset\}$, we have:

$$g(\boldsymbol{f}_0 + \sum_{k \in \Omega_S} \Delta \boldsymbol{f}_k) = \mathbf{x}_0 + \sum_{k \in \Omega_S} \Delta \mathbf{x}_k$$

Since other feature components whose action field do not cover $S$, they have no generation effects on patches in $S$. Therefore, the superposition of feature components in $\Omega_S$ is sufficient for the generation of patches in $S$, i.e.:

$$\forall S \subseteq N, \ \mathrm{region}(S \mid g(\boldsymbol{f}')) = \mathrm{region}(S \mid \mathbf{x}) \quad s.t. \quad \boldsymbol{f}' = \boldsymbol{f}_0 + \sum\nolimits_{k \in M: A_k \cap S \neq \emptyset} \Delta \boldsymbol{f}_k,$$

which is Theorem 1-$\alpha$ in the main paper. $\square$

**Proving that Theorem 1-$\alpha$ could be rewritten as Theorem 1-$\beta$.**

*Proof.* According to Theorem 1-$\alpha$, reconstructing patches in $S \subseteq N$ equals summing up feature components in $\Omega_S = \{k \mid A_k \cap S \neq \emptyset\}$. From another perspective, whether a feature component $\Delta \boldsymbol{f}_k$ should be added to the intermediate feature depends on the reconstruction demand: if the action field $A_k$ of the feature component $\Delta \boldsymbol{f}_k$ cover (or partially cover) the reconstruct target $S$, then feature component $\Delta \boldsymbol{f}_k$ should be added for the reconstruction.

Therefore, given a demand of generating a set of image patches $S \subseteq N$, the feature component $\Delta \boldsymbol{f}_k$ with an action field $A_k = \{i_1, i_2, \ldots, i_l\}$ must be added to the baseline feature $\boldsymbol{f}_0$, if any patch $i_1$, **or** $i_2$, **or**, $\ldots$, **or** $i_l$ is contained within $S$. This is the Theorem 1-$\beta$ in the main paper, which explicitly indicates the OR relation between patches in action fields of the feature components. $\square$

**Proving that feature components extracted using OR interaction satisfy linear additivity and feature completeness**.

*Proof.* We begin by proving the OR-interaction-based feature components satisfy the linear additivity property, as demonstrated in Theorem 2 of the main text.

According to the definition of the Harsanyi interaction, we have $\forall S \subseteq N$,

$$\sum_{k:A_k \cap S \neq \emptyset} \Delta \boldsymbol{f}_k = - \sum_{k:A_k \cap S \neq \emptyset} \sum_{L \subseteq A_k} (-1)^{|A_k|-|L|} \hat{\boldsymbol{f}}_{N \setminus L}$$

$$= \sum_{L \subseteq N} \sum_{k:A_k \cap S \neq \emptyset, A_k \supseteq L} (-1)^{|A_k|-|L|} \hat{\boldsymbol{f}}_{N \setminus L}$$

$$= \sum_{L \subseteq N} \sum_{t=|L|}^{|N|} \sum_{\substack{k:A_k \cap S \neq \emptyset, A_k \supseteq L \\ |A_k|=t}} (-1)^{t-|L|} \hat{\boldsymbol{f}}_{N \setminus L}$$

$$= \sum_{L \subseteq N} \hat{\boldsymbol{f}}_{N \setminus L} \sum_{m=0}^{|N|-|L|} (\binom{|N|-|L|}{m} - \binom{|N|-|L \cup S|}{m})(-1)^m$$

$$= \hat{\boldsymbol{f}}_S$$

Therefore, we have $\hat{\boldsymbol{f}}_S = \sum_{k:A_k \cap S \neq \emptyset} \Delta \boldsymbol{f}_k$ (linear additivity, Theorem 1-$\alpha$).

From another perspective, given a demand of generating a set of image patches $S \subseteq N$, the feature component $\Delta \boldsymbol{f}_k$ with an action field $A_k = \{i_1, i_2, \ldots, i_l\}$ must be added to the baseline feature $\boldsymbol{f}_0$ to reconstruct $\hat{\boldsymbol{f}}_S$, if any patch $i_1$, **or** $i_2$, **or**, $\ldots$, **or** $i_l$ is contained within $S$. Therefore, the extracted feature components also satisfy Theorem 1-$\beta$.

Specifically, when set $S = N$, we have $\sum_{k \in M} \Delta \boldsymbol{f}_k = \hat{\boldsymbol{f}}_N = \boldsymbol{f}$. Therefore, the feature components extracted using OR interactions satisfy feature completeness (Axiom 1). $\square$

**Proof for Theorem 2**.

*Proof.* Based on the above reasoning, we can now establish Theorem 2. Given a logical model $h(S) = \boldsymbol{f}_0 + \sum_{k \in M} \mathbb{1}(i_1 \in A_k \text{ or } i_2 \in A_k, \ldots, \text{ or } i_l \in A_k) \cdot \Delta \boldsymbol{f}_k$, we have:

$$h(S) = \boldsymbol{f}_0 + \sum_{k \in M} \mathbb{1}(i_1 \in A_k \text{ or } i_2 \in A_k, \ldots, \text{ or } i_l \in A_k) \cdot \Delta \boldsymbol{f}_k$$

$$= \boldsymbol{f}_0 + \sum_{k:A_k \cap S \neq \emptyset} \Delta \boldsymbol{f}_k$$

$$= \hat{\boldsymbol{f}}_S$$

Then, based on the definition of minimal features in Equation (5), we have:

$$\| \text{region}(S \mid \tilde{\mathbf{x}}) - \text{region}(S \mid \mathbf{x}) \|_2 < \epsilon \quad \text{subject to} \quad \tilde{\mathbf{x}} = g(h(S))$$

$\square$

# E  IMPLEMENTATION DETAILS

The computation of the minimal feature $\hat{\boldsymbol{f}}_S$ *w.r.t.* image regions in $S$ can be approximated as:

$$\min_{\alpha} \|\boldsymbol{f}_S\|_{L-1} + \frac{\lambda}{|S|} \| \text{region}(S|\tilde{\mathbf{x}}) - \text{region}(S|\mathbf{x}) \|_2,$$

$$\textit{subject to } \| \text{region}(S|\tilde{\mathbf{x}}) - \text{region}(S|\mathbf{x}) \|_2 < \epsilon,$$

$$\boldsymbol{f}_S = \boldsymbol{f}_0 + \alpha \odot \Delta \boldsymbol{f} \tag{9}$$

Here, we consider the minimal feature are all *contained* by the whole feature $\boldsymbol{f}$, so we apply the empirical constrain $\hat{\boldsymbol{f}}_S = \boldsymbol{f}_0 + \boldsymbol{\alpha} \odot (\boldsymbol{f} - \boldsymbol{f}_0), w.r.t. \, \boldsymbol{\alpha} \in [0,1]^D$. $\epsilon$ is a small scalar threshold, which bounded the reconstruction error. In our experiment, $\epsilon$ is set as $10^{-4} * number\ of\ pixels\ in\ S * number\ of\ channels$. Each feature dimension of $\boldsymbol{\alpha}$ is in the range of [0,1]. $\odot$ is referred to as the element-wise multiplication. In this way, we constrain $\boldsymbol{f}_S$ strictly locates within the range between $\boldsymbol{f}_0$ and $\boldsymbol{f}$.

This optimization problem can be solved using gradient descend method. *i.e.,*

$$\boldsymbol{\alpha}_{t+1} = \boldsymbol{\alpha}_t - \eta \cdot \nabla_{\boldsymbol{\alpha}} \mathcal{L}(\boldsymbol{\alpha}_t) \quad s.t. \quad \boldsymbol{\alpha}_0 = \boldsymbol{0} \tag{10}$$

where $\boldsymbol{0}$ denotes a tensor with all elements 0, $\mathcal{L} = \|\boldsymbol{f}_S\|_{L-1} + \frac{\lambda}{|S|} \| \text{region}(S|\tilde{\mathbf{x}}) - \text{region}(S|\mathbf{x}) \|^2$.

Table 2 shows the hyperparameters of this optimization problem *w.r.t.* different neural network architectures.

Table 2: The hyperparameters of optimizing minimal features *w.r.t.* different neural network architectures

| network | learning rate | $\lambda$ | iterations |
|---|---|---|---|
| BigGAN | 1e-4 | 1e3 | 500 |
| StyleGAN | 1e-3 | 1e4 | 500 |
| NVAE | 1e-3 | 1e4 | 200 |
| SiD Diffusion Model | 1e-3 | 1e4 | 200 |

**Implementation details of verifying the linear additivity property.** We ranked the feature components by their L2-norm, then selected the top-30 of all the 512 feature components as the most salient feature components. We followed Equation (7) to mimic the minimal feature using only salient feature components.

## F CONTROLLABLE IMAGE GENERATION BASED ON FEATURE COMPONENTS

According to Theorem 1-$\alpha$, we could effectively reconstruct patches in a set $S$, if all feature components, whose action fields (partially) covered the target patch set $S$, were added to the baseline feature $\boldsymbol{f}_0$. *I.e.*, we can obtain the minimal feature $\hat{\boldsymbol{f}}_S$.

Therefore, we conducted experiments to achieve controllable image generation based on feature components. We followed the experimental settings in Section 3 to segment the input image into different patches. First, a set of patches $S \subseteq N$ were randomly selected from the image $\mathbf{x}$. Let $\hat{\Omega} = \{k | A_k \cap S \neq \emptyset\}$ denote a set of feature components whose action field (partially) covered the target patch set $S$. Then, we incrementally added more feature components in $\hat{\Omega}$ to the baseline feature $\boldsymbol{f}_0$. We use the RMSE $= \sqrt{\| \text{region}(S|\tilde{\mathbf{x}}) - \text{region}(S|\mathbf{x}) \|^2 / (|S| \cdot L)}$ to evaluate the reconstruction error, where $\text{region}(S|\tilde{\mathbf{x}})$ denotes an operation function that clips patches in the set $S$ from the reconstructed image $\tilde{\mathbf{x}}$, $|S|$ denotes the number of patches in $S$, and $L$ denotes the number of pixels in a patch of image $\mathbf{x}$.

As shown in Figures 9 and 10, the target image patch set was progressively reconstructed when we kept adding more feature components in $\hat{\Omega}$. Ultimately, by adding feature components from $\hat{\Omega}$, the target patch set was gradually reconstructed. The small reconstruction error verified that the extracted feature components enabled controllable generation.

Figures 9 and 10 also show that patches outside the target patch set were only partially reconstructed and not fully reconstructed. This could also be explained by the theory proposed in this paper. When we gradually added feature components in $\hat{\Omega}$, we only required the target set of patches to be accurately reconstructed, with no specific requirement on the reconstruction quality of regions outside the target set of patches. Since the action fields of the added feature components usually also covered other image patches that were not in the target patch set, image patches outside the target set were partially reconstructed.

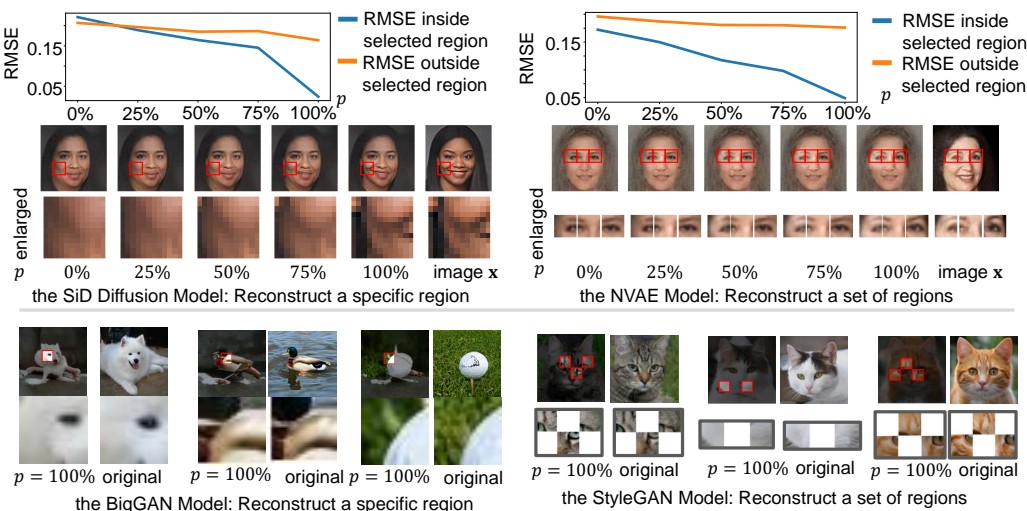

Figure 9: Validation of controllable generation, *i.e.*, relevant feature components reconstruct the target region. We added different ratios $p$ of feature components whose action regions covered the target image regions (in red boxes). We found that the target image region was gradually reconstructed when we added increasing number of feature components.

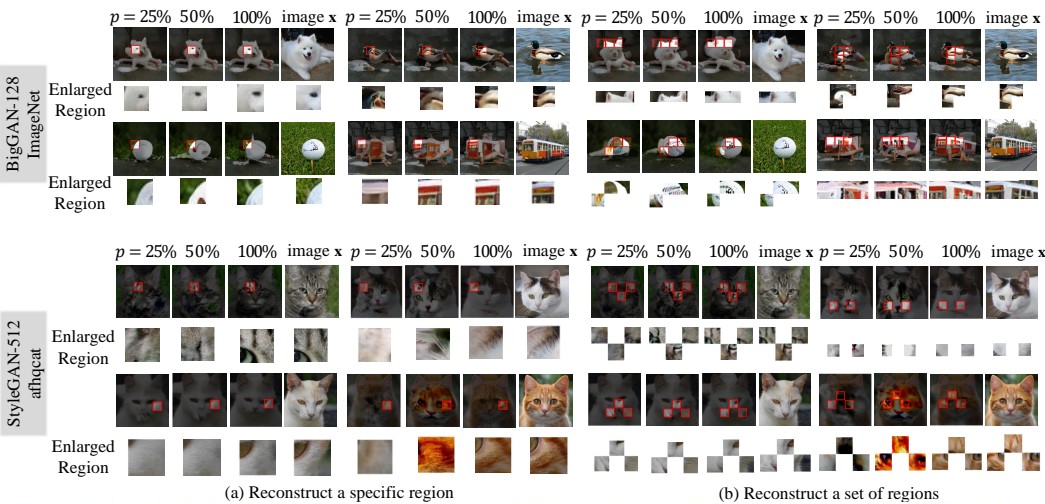

(a) Reconstruct a specific region
(b) Reconstruct a set of regions

Figure 10: Additional results on BigGAN and StyleGAN for controllable generation validation.

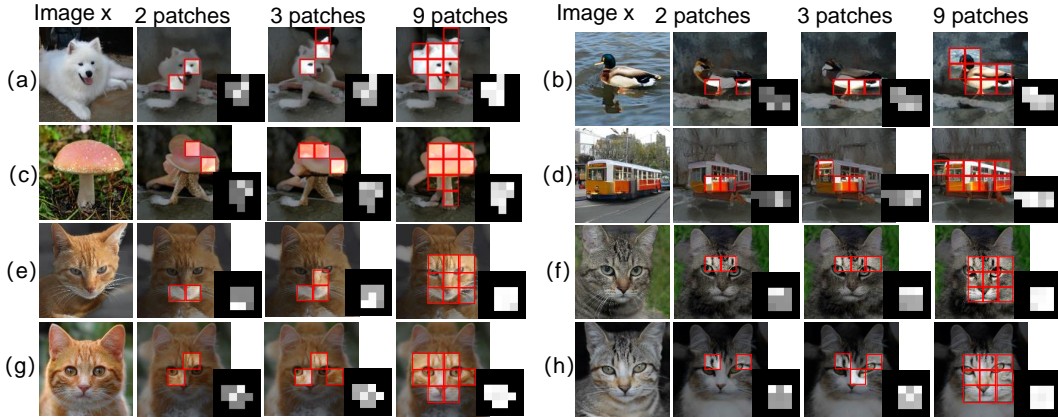

Figure 11: Incremental reconstruction of different image regions when we gradually added feature components. We added all feature components corresponding to the target image regions (in red boxes) for image reconstruction. It shows that different regions in the target object were sequentially reconstructed, but these feature components did not reconstruct the background. The heatmap shows the distribution of the overlapping action regions of the added feature components.

**Sequentially reconstructing image patches**. Theoretically, if feature components are faithfully disentangled, we can use such feature components to accurately control the image generation process, *e.g.*, sequentially reconstructing image patches one-by-one. Therefore, we conducted experiments to examine the quality of the controlled sequential reconstruction of image patches, so as to evaluate the faithfulness of our method.

We simply followed the experimental settings in Section 3, and sequentially reconstructed image patches, as follows. In each step, we randomly selected a new patch, and added all feature components that covered the target new patch to reconstruct this new patch. By doing this, all image patches were supposed to be sequentially reconstructed one-by-one.

Figure 11 and 12 shows the experimental result of sequentially reconstructing image patches one-by-one. In each step, the newly added patch was accurately reconstructed. Note that when the DNN well reconstructed a set of image patches, the other patches were also partially influenced, but these patches were not fully reconstructed. This could also be explained by our theory, because these influenced patches were also covered by the added feature components' action fields. Figure 11 illustrates the quality of the sequential reconstruction of image patches, which verified the faithfulness of the disentangled feature components.

# G   MORE EXPERIMENT RESULTS

This subsection presents additional results from the experiments in the main paper.

## G.1   MORE RESULTS ON VISUALIZING REGIONAL PATTERNS

Following the same Experiment setting in Section 2.2, Figure 13 shows more visualization results of regional patterns on the NVAE model.

## G.2   MORE RESULTS ON VERIFYING THE SPATIAL BOUNDEDNESS PROPERTY

We conducted experiment on more models and samples to verify the spatial boundedness property. We followed the experiment setting in Section 3.2. Figure 14 and 15 shows results on the SiD Diffusion, BigGAN, and StyleGAN models. We observed that the spatial boundedness property held across different models.

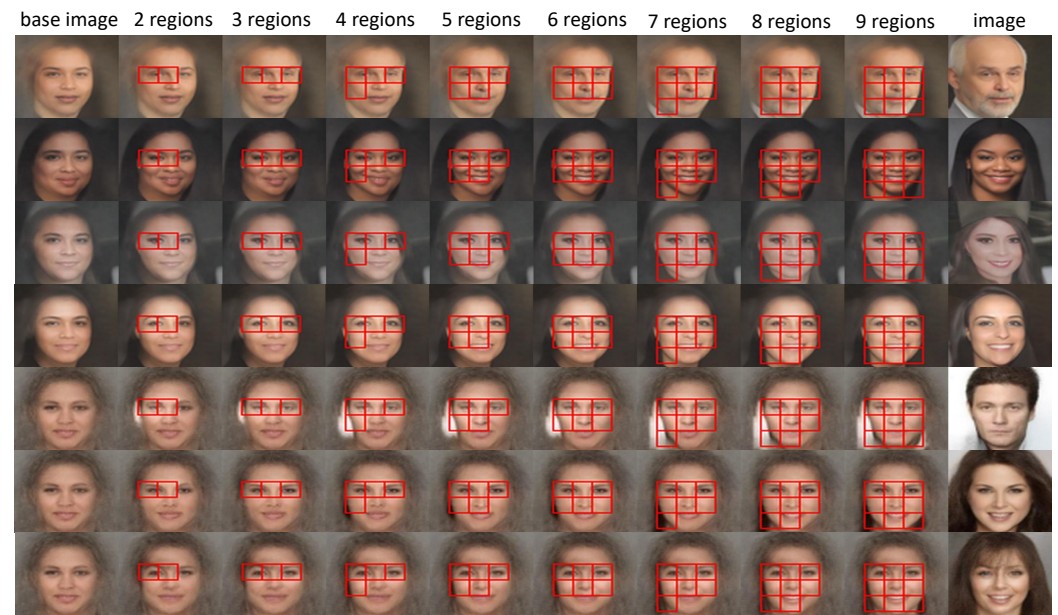

Figure 12: Additional results on the SiD Diffusion model and the NVAE model for incremental reconstruction of different image regions when we gradually added feature components. The first four rows were from experiments using the SiD Diffusion model, and the last three were from the NVAE model.

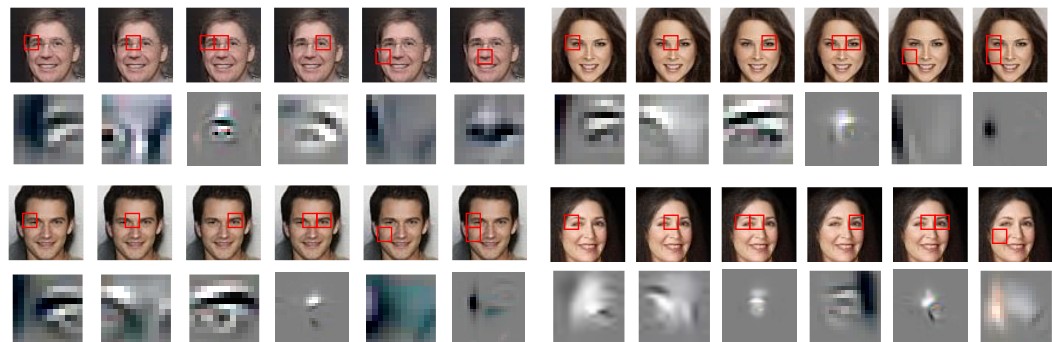

Figure 13: More results on visualizing regional patterns on NVAE.

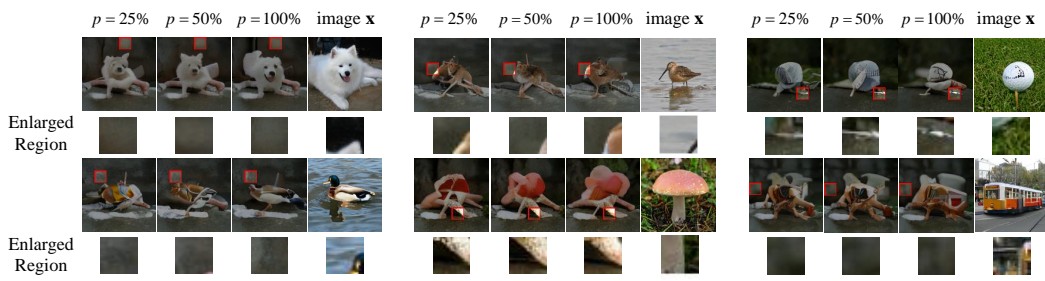

Figure 14: More results conducted on the BigGAN model to test whether the target patch $i$ was not affected by all feature components whose action fields did not cover the patch.

(a) γ=0.994 γ=0.971 γ=0.987

(b) γ=0.989 γ=0.986 γ=0.994

(c) γ=0.999 γ=0.999 γ=0.993

(d) γ=0.994 γ=0.999 γ=0.995

(e) γ=0.990 γ=0.995 γ=0.986 γ=0.942

(f) γ=0.984 γ=0.994 γ=0.993 γ=0.983

(g) γ=0.967 γ=0.990 γ=0.997 γ=0.985

(h) γ=0.971 γ=0.990 γ=0.993 γ=0.990

Figure 15: Additional results to verify the spatial boundedness property, with samples from (a,b) BigGAN, (c,d) StyleGAN, and (e,f) SiD Diffusion models.

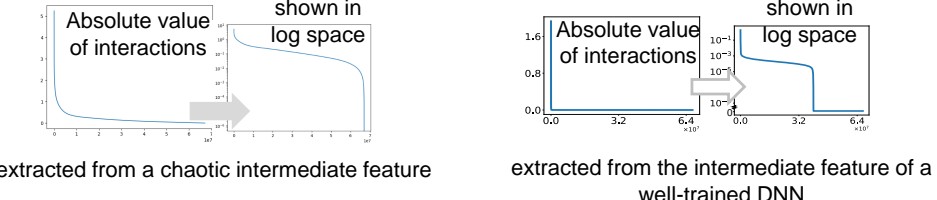

Figure 16: An example of dense feature components. The feature components extracted from chaotic intermediate feature(shown in left figure) were more dense than the feature components extracted from the feature components of a well-trained DNN(shown in right figure). Such result indicated that not all DNNs encode sparse feature components.

## H  DETAILED DISCUSSION ON THE DEFINITION OF SPARSITY

Originally, the sparsity is defined for scalar interaction values, *i.e.*, describing a state that almost all interactions have negligible values, and only very few interactions have salient values $I_{or}(S)$. This definition has been widely used by Cheng et al. (2024), Zhang et al. (2024) and Zhou et al. (2023) in the scope of interaction-based explanation. However, because the OR interaction in this study is a vector, rather than a scalar, we define the sparsity at the level of elements of the feature component $\Delta f_k$.

## I  DISCUSSION ON WHEN WILL DNN ENCODES DENSE FEATURE COMPONENT

If a DNN encodes chaotic representation or randomly generates fully noisy outputs, then the extracted feature components would be very dense, *i.e.,* each feature component would contain dense and chaotic neural activations.

We randomly initialize the intermediate feature $f$ of StyleGAN as a random tensor. Then, based on this chaotic intermediate feature, we further applied our algorithm. Figure 16 shows that the extracted feature components were dense, comparing with the sparse feature components in Figure 3. Such dense feature component can not be viewed as a clear representation of the DNN. Therefore, if a DNN encodes chaotic representation or randomly generates fully noise outputs, which leads to chaotic intermediate feature, the extracted feature components would be very dense.

## J  EXPERIMENTS COMPUTE RESOURCES AND COMPUTATIONAL ANALYSIS

**Experiments Compute Resources.** In our algorithm, the computationally intensive segment predominantly lies in the optimization of minimal features. Our experiments were conducted utilizing four 3080Ti GPUs. For the NVAE, BigGAN, StyleGAN, and SiD Diffusion models, the computation of all 512 minimal features approximates a duration of 30 minutes. However, by employing some commonly used acceleration techniques, such as half-precision training and multi-process/multi-thread technologies, the efficiency of the algorithm proposed in this paper can be significantly improved.

**Computational Complexity and Acceleration Strategies**: The core step of our algorithm involves the optimization of the minimal feature $\hat{f}_S$ for each subset $S$ of the target patch set $N$. Although the theoretical time complexity for this step is $O(2^{|N|})$, we can apply a strategy to effectively prevent the exponential increase in computational load.

• Method: We partition the image into $K$ overlapping local groups of size $n'$ (where $n' \ll |N|$). This reduces the complexity from exponential $O(2^{|N|})$ to linear-scale $O(K \cdot 2^{n'})$.

• Empirical Validation: With $n' = 9$ and multi-process parallelization, we can compute the structure for a full image in 30 minutes. This proves our method can scale beyond small patch subsets to full-image analysis.

## K  VERIFYING GENERALITY ON ARBITRARY PATCH SETS

To demonstrate that the proposed representation structure is general and not limited to specific image regions (e.g., foreground objects), we conducted additional experiments on a new patch set $N'$ randomly selected from the entire image, including background areas.

Given an image $x$ generated by BigGAN, we selected an arbitrary patch set $N'$, distinct from the foreground-focused set $N$ analyzed in the main text. We evaluated the faithfulness of the representation structure corresponding to $N'$ using the two key metrics defined in Section 3:

• Spatial Boundedness ($\gamma$): To verify whether the regional patterns $\Delta x_k$ are spatially confined within their action fields $A_k$, we utilized the metric $\gamma = \|\text{region}(A_k \mid \Delta x_k)\|_2 / \|\Delta x_k\|_2$ (proposed in Paragraph 1 of Section 3.2). A higher $\gamma$ (closer to 1.0) indicates that the regional pattern is predominantly contained within the action field of $\Delta f_k$.

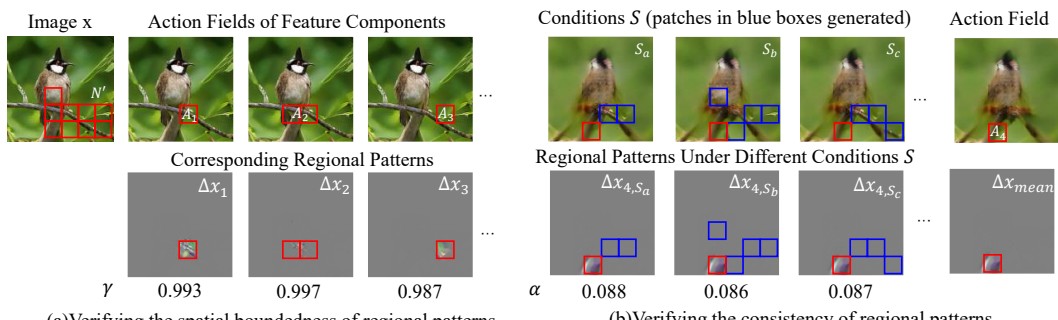

(a)Verifying the spatial boundedness of regional patterns    (b)Verifying the consistency of regional patterns

Figure 17: Verification of **(a)** spatial boundedness and **(b)** consistency of the representation struc-ture corresponding to the newly selected patches $N$. The experiment is conducted on a BigGAN-generated image $x$, where red bounding boxes indicate the action fields $A_k$ (corresponding to the newly selected patches $N$). (a) Regional patterns $\Delta x_k$ generated by feature components $\Delta f_k$ demon-strate strict spatial localization, quantified by the high relative strength $\gamma$ within the action field $A_k$. (b) Regional patterns $\Delta \mathbf{x}_{k,S}$ under varying contexts $S$ (blue bounding boxes) showed consistent vi-sual appearance, quantified by the low inconsistency scores $\alpha$.

• Consistency ($\alpha$): To verify whether $\Delta x_k$ remains consistent under varying surrounding contexts $S$, we utilized the metric $\alpha = \|\Delta x_{k,S} - \Delta x_{\mathrm{mean}}\|_2 / \|\Delta x_{\mathrm{mean}}\|_2$ (proposed in Paragraph 1 of Section 3.3). Here, $\Delta x_{k,S}$ denotes the pattern generated by $\Delta f_k$ when conditioned on a context $S$ outside the action field ($S \cap A_k = \emptyset$). A lower $\alpha$ indicates that the pattern is stable and robust to contextual changes.

Experimental results based on the newly selected patches $N$ are shown in Figure 17:

• Spatial Boundedness: All regional patterns $\Delta x_k$ exhibited $\gamma$ values greater than 0.98, indicating they were predominantly contained within their respective action fields $A_k$ (Fig. 17(a)). The results demonstrated that the representation structure corresponding to the newly selected patches in $N$ maintained strong spatial boundedness.

• Consistency: The $\alpha$ values of the regional patterns $\Delta x_k$ were smaller than 0.09. The low $\alpha$ values confirmed that the feature component $\Delta f_k$ consistently generated almost same regional patterns $\{\Delta x_{k,S}\}$ even when the surrounding context $S$ changed (Fig. 17(b)). Results validated that the representation structure corresponding to $N$ maintained strong consistency.

These results confirmed that the key properties (spatial boundedness and consistency) held for arbi-trary sets of patches, demonstrating that our method was generally applicable and not limited to a biased subset (e.g., foreground) of the generated image.

## L  SYMMETRIC EVALUATION OF TRANSFERABILITY

We have redesigned the metric to ensure a symmetric comparison for transferability, as follows:

Specifically, given a pair of generated images $(\mathbf{x}^{(a)}, \mathbf{x}^{(b)})$ containing the same object, we selected the same number (top-$l$) of feature components with the highest L2-norm from both images. Let $D_{\mathbf{x}^{(a)}}$ and $D_{\mathbf{x}^{(b)}}$ denote the sets of the corresponding action fields, respectively.

We then quantified transferability using the transferability score $t = |D_{\mathbf{x}^{(a)}} \cap D_{\mathbf{x}^{(b)}}|/l$. This metric measures the proportion of feature components extracted from $\mathbf{x}^{(a)}$ that could be transferred to the representation of image $\mathbf{x}^{(b)}$. By fixing the number of selected components ($l$) for both sides, this new metric guarantees a fair and symmetric assessment.

Experiments on NVAE-generated image pairs, as detailed in Figure 18, showed that under this sym-metric metric, the transferability score $t$ was 0.85. This high $t$ value confirmed that a significant majority of salient regional patterns were indeed effectively transferred across images of the same object.

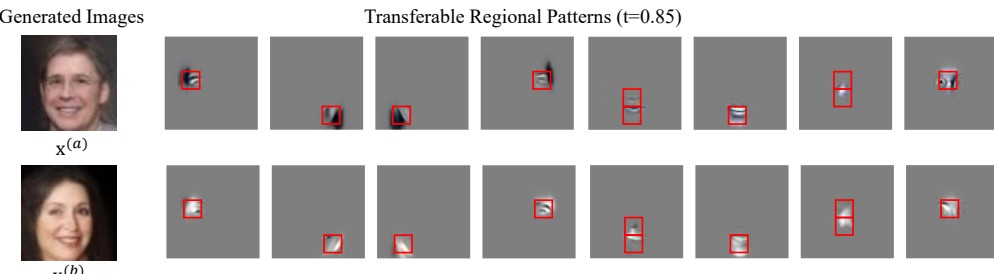

Figure 18: Visualization of transferable regional patterns in NVAE. The first column shows distinct samples $x^{(a)}$ and $x^{(b)}$, followed by transferable patterns with action fields highlighted in red boxes. The high transferability score ($t = 0.85$) confirmed that salient patterns maintained semantic consistency across identities, exhibiting only subtle contextual variations. Notably, we observed that transferable regional patterns tend to be low-order (i.e., covering a small number of image patches).

## M    LIMITATIONS

In this study, minimal features are calculated in an engineering manner, resulting in high time complexity and limited precision. As a preliminary exploration of DNNs' internal representation for image generation, our method relies on the common setting of using image grids as input (Ren et al., 2023a; Li & Zhang, 2023; Ren et al., 2024a). Future work will focus on automatically segmenting compositional image patches to obtain sparser feature components.

## N    BROADER IMPACTS

This paper presents a new method to explain the internal representation of image generation conducted by neural networks. The image generation can be explained as the superposition of primitive regional patterns. Each primitive regional pattern is generated by a feature component disentangled from a intermediate feature. Such feature components are computed by OR Harsanyi interaction, which was originally used to explain classification model. This paper extends the usage of OR interaction in the field of explainable AI.

## O    THE USE OF LARGE LANGUAGE MODELS (LLMS)

This paper employs large language models (LLMs) solely for partial polishing of wording and phrasing, with the aim of enhancing the clarity and standardization of the writing.

