# OpenReview forum: "Disentangling Primitive Representation Structures for Image Generation"
_ICLR.cc/2026/Conference — Submitted to ICLR 2026_

### Official Review · Reviewer_uSg2 · 2025-10-24

**Soundness:** 3
**Presentation:** 2
**Contribution:** 2
**Rating:** 4
**Confidence:** 2

**Summary:**

This paper proposes a method for interpretation of generative models.
Namely, the authors study the outputs of a middle layer of a generative network, and try to decompose it into so-called primitive patterns.
Those patterns should satisfy a set of axioms, also proposed by the authors.
The experimental section demonstrates that indeed these properties can be reasonably fulfilled, and the resulting image can be thought of as a composition of low-level patterns.

**Strengths:**

I am not an expert in interpretability of generative models, and for me the presented ideas sound quite novel and interesting.
The proposed set of desirable properties looks well-motivated, and the experimental results indeed demonstrate that some meaningful patters can be found in this way.
Probably, this approach can be interesting to the members of ICLR community that work in the field of explainable AI.

**Weaknesses:**

1. From my personal perspective, usefulness of such decomposition can be measured by how well it solves the problem of image editing. The prior approaches, e.g. [1] and [2], criticized by the authors, allowed such semantic editing. And, in my opinion, their editing results were more convincing than the ones presented in this manuscript (Figs. 8 and 10).

2. The whole procedure of learning feature components is not described very well in the main text. While they are defined through their action fields (Eq. 6), it seems from lines 900-901, that subsets of patches are selected randomly. It is unclear, how this randomness affects the learned feature components. Isn't it natural to assume that different selection of the patches will lead to different components? And is there any possibility to identify the best  action fields? Overall, I find this part of the paper very unclear.


References.

[1] A. Voynov and A.Babenko. Unsupervised discovery of interpretable directions in the GAN latent space. In ICML, 2020.

[2] Z. Chen et al. Exploring gradient-based multi-directional controls in gans. In ECCV, 2022.

**Questions:**

I ask the authors to address the weaknesses listed above:

1. Explain more clearly how action fields are selected, and how feature components are learned afterwards.
2. Provide meaningful examples of image editing by means of learned feature components.

---

> ### Author Response · Authors · 2025-11-22
> **Rebuttal By Authors**
>
> ## Response
>
> Thank you. Please find our responses to your questions below. For questions that remain unanswered, we will respond in a few hours.
>
> $\colorbox{#FFF9C4}{W1: “Usefulness of such decomposition can be measured by how well it solves the problem of image editing”}$
>
> $\colorbox{#FFF9C4}{Q2: Could the authors “provide meaningful examples of image editing by means of learned feature components”?}$
>
> This is a very interesting question. In fact, "objective mechanistic explanation" and "pragmatic engineering techniques" have always been two conflicting schools of thought in the field of explainable AI. From a pragmatic perspective, image editing is naturally a widely-watched problem. However, from the perspective of developing fundamental theories in explainable AI, how to objectively explain the representation structure modeled by neural networks in image generation is the biggest challenge for interpretability research on generative networks, and it remains a theoretical void.
>
> In this direction, previous work has only been able to explain the internal representation mechanisms of classification decision models (such as those with a softmax function for classification), for example, using Shapley values or interaction-based explanations for large language models. A significant body of research focuses on mechanistic explanations, such as visualizing the visual or linguistic features modeled in the intermediate layers of neural networks (e.g., Microscope [1] by OpenAI, Deep Dreaming with BERT [2] by Google). However, we are the first to extract a concise and interpretable representation structure, which encompasses all feature components, to precisely explain how a neural network generates an image. This is a completely new problem. More crucially, we have experimentally discovered the sparsity of feature components representing an image and the transferability of regional patterns across different images, which indicates that the internal representation structure of image generation networks extracted by our method is rigorous and reliable.
>
> In summary, the root of this question lies in the conflict between different schools of interpretability, and our work plays a significant role in exploring the fundamental mechanisms of image generation networks. Our method can also be used for image editing; for example, adding a feature component $\Delta f_k$ to the intermediate feature $f$ can produce the corresponding regional pattern $\Delta x_k$ on the image. Please see Figs. 9,11,12 in our paper for details. However, our method is not exclusively designed for image editing.
>
> Reference:
>
> [1] OpenAI. (2020, April 14). Microscope. [https://openai.com/index/microscope/](https://openai.com/index/microscope/)
>
> [2] Bäuerle, A., & Wexler, J. (n.d.). Deep dreaming with BERT. People + AI Research (PAIR), Google. [https://pair-code.github.io/interpretability/text-dream/blogpost/](https://pair-code.github.io/interpretability/text-dream/blogpost/)

---

> ### Author Response · Authors · 2025-12-02
> **Rebuttal to Reviewer uSg2**
>
> ## Response
>
> $\colorbox{#FFF9C4}{W2:“Isn't it natural to assume that a different selection of the patches will lead to different components? And is there any possibility to identify the best action fields?”;}$
> $\colorbox{#FFF9C4}{Q1: Could the authors “explain more clearly how action fields are selected, and how feature components are learned afterwards”?}$
>
> Thank you. We followed your concerns to conduct **a new experiment** on patches randomly selected from the entire image. It is natural that different patch selections lead to different feature components, as the components are designed to encode specific regional information. However, one of the core contributions of our method is that regardless of which patches are selected, the learned representation structure consistently maintains the key properties: spatial boundedness and consistency.
>
> To empirically verify that our method effectively identifies valid representation structures for arbitrary patch selections (rather than being limited to a specific "best" set or foreground regions), we randomly selected a new patch set $N'$ covering various random image regions (different from the foreground-focused patch set $N$ discussed in the main paper) and evaluated the faithfulness of the learned representation structure using our proposed metrics:
>
> **Metrics**:
>
> - To verify whether the extracted regional patterns $\Delta x_k$ are spatially located in its action fields $A_k$, we used the metric $\gamma= \Vert region(A_k \mid \Delta x_k)\Vert_2/\Vert \Delta x_k \Vert_2 $ (proposed in Paragraph 1 of Section 3.2.1, Line 362) to validate the spatial boundedness property. The metric $\gamma$ evaluates the strength of the regional pattern Δx_k within the action field A_k relative to its total strength ||Δx_k||. A higher $\gamma$ means the regional pattern Δx_k is predominantly contained within $\Delta f_k$’s action field A_k，indicating the regional patterns satisfy the spatial boundedness property.
>
> - To verify whether the extracted regional patterns $\Delta x_k$ is consistent under different surrounding contexts $S$, we used the metric $\alpha= \Vert \Delta x_{k,S} - \Delta x_{mean} \Vert_2/\Vert \Delta x_{mean}\Vert_2 $ （proposed in Paragraph 1 of Section 3.3, Line 417）to validate the consistency property. The metric $\alpha$ evaluates the relative deviation of the regional patterns $\{\Delta x_{k,S}\}$ generated by $\Delta f_k$ conditioned on varying patch sets $S$ outside the action field $A_k$, i.e., $S \cap A_k= \emptyset$. A lower $\alpha$ value indicates that the regional patterns $\{\Delta x_{k,S}\}$ under different contexts $S$ remain consistently close to the mean regional pattern $\Delta x_{k,mean}$, demonstrating that the regional patterns satisfy the consistency property.
>
> **Experimental results**: Results based on the newly selected patches $N’$ are shown in Figure 17 of Appendix K of the revised paper:
>
> - Spatial Boundedness: All regional patterns $\Delta x_k$ exhibited $\gamma$ values greater than 0.98, indicating they were predominantly contained within their respective action fields $A_k$ (Fig. 17(a)). The results demonstrated that the representation structure corresponding to the newly selected patches in $N’$ maintained strong spatial boundedness.
>
> - Consistency: The $\alpha$ values of the regional patterns $\Delta x_k$ were smaller than 0.09. The low $\alpha$ values confirmed that the feature component $\Delta f_k$ consistently generated almost same regional patterns $\{\Delta x_{k,S}\}$ even when the surrounding context $S$ changed (Fig. 17(b)). Results validated that the representation structure corresponding to $N’$ maintained strong consistency.
>
> These results confirmed that the key properties (spatial boundedness and consistency) held for arbitrary sets of patches, demonstrating that our method was generally applicable and not limited to a biased subset (e.g., foreground) of the generated image.

---

### Official Review · Reviewer_7rhP · 2025-10-30

**Soundness:** 2
**Presentation:** 3
**Contribution:** 2
**Rating:** 4
**Confidence:** 3

**Summary:**

This paper completely breaks down one whole feature in the middle hierarchy of deep neural network DNNs and defines a representative structure that can interpret image generation as a simple sum of multiple primitive feature components.
The core of this structure is that each primitive feature component has its own action field and is only responsible for generating image patterns in that particular region.
The author first establishes a rigorous foundation by presenting three axioms that a 'Faithful' explanation of a model's internal workings must satisfy: Feature Completeness, Spatial Boundness, and Consistency.
Then, to extract independent components that satisfy these axioms, they introduce a mathematical formula called OR interaction.
The predictive features of components extracted from four different models (SiD Diffusion, NVAE, BigGAN, StyleGAN) using OR interaction successfully demonstrated all three axioms: they showed linear additionality matching the real minimum features, remained spatially bound as patterns did not leak outside their Action Field, and generated consistent patterns even when conditions changed.

**Strengths:**

Strong points
This paper proves that the proposed methodology is mathematically satisfactory and visually and numerically demonstrates it through systematic experimental verification.
Sparsity finds that most of the extracted feature components are zero and only a small number of values are activated, and that only a small number of components contribute to image generation.
When model draws different faces from each other, they find that they reuse the eye and nose feature components’ regional patterns they used before.
It uses components to create precise control over only certain areas of the image in the order in which it is desired.

**Weaknesses:**

Weak points
This paper argued for the integrity of being able to completely decompose the entire image (e.g., 36 patches), but due to the computational cost problem of NP-complete, the actual experiment only takes place in a reduced environment of any 9 patches (1/4). It's okay if it's 9 patches, but I wonder if I can guarantee that it's okay if it's 36 patches (whole).
To verify transferability, the Top 20 transferable regional patterns list in Image A was compared with the Top 100 transferable regional patterns list in Image B. Fair comparison is Top 20 vs. Top 20 or Top 100 vs. Must be Top 100. The author does not provide an academic basis for any number l_a=20 or l_b=100. As a result, the high t value of 76-88% may have been intentionally inflated, and the results of this experiment are unreliable.

Main argument
(1)
The experimental verification presented in this paper is insufficient to support the validity of the proposed methodology.
3 EXPERIMENT - Narrow Patch Selection
The data processing method used for internal representation structure analysis (the process of dividing and selecting image patches) introduces limitations that seriously hinder the generalizability of the methodology. The author divides the generated image x into a uniform 6 ×6 grid, defining a total of 36 patches. This partitioning is an essential preprocessing step to specifically understand what Action Fields a particular Regional Pattern is associated with; that is, which feature components affect which part of the image. However, the core experiment did not cover all 36 patches. The authors cited Li & Zhang (2023) and analyzed only n=9 patches selecting them randomly from the foreground object.
While this approach is understandably due to the enormous computational cost of computing interactions, an analysis based solely on 9 foreground patches cannot fully explain the complex representation structure of the entire 36 patch grid. Therefore, the current experimental results can only be considered valid for a biased subset of the model's output (i.e., the foreground). This remains a significant limitation, as it is insufficient to verify the general representation structure claimed by the paper.

3.4 TRANSFERABILITY OF FEATURE COMPONENTS’ REGIONAL PATTERNS
This paper compares the list of feature components in the top $l_a=20$ from image $x^((a) )$ with the list in the top $l_b=100$ from image $x^{(b)}$ to verify the transferability ratio ($t$). Such an asymmetric comparison ($l_a≠l_b$) raises serious questions about the fairness of the experiment.
A fair comparison of importance between the two sets would require a symmetric approach, such as Top 20 vs. Top 20 or Top 100 vs. Top 100. Furthermore, the authors provide no academic or experimental rationale (or justification) for why they chose these specific asymmetric values of $l_a=20$ and $l_b=100$. The absence of such justification does not rule out the possibility that the reported $t$ values (76-88%) were intentionally overestimated. The current setup means that a feature component that was, for example, the 15th most important in $x^((a) )$ would still be counted as transferred even if it were only the 95th most important (i.e., almost meaningless) in $x^((b) )$. This distorts the answer to the more critical question, "Is the core pattern still core in other images?", and seriously undermines the reliability of the results. Therefore, the authors must provide a clear rationale for this asymmetric setting or, preferably, re-evaluate the t value by conducting a sensitivity analysis using a fair, symmetric comparison (e.g., $l_a=l_b$).

(2)
Line 358: The pixel-wise differences of the pattern $∆x_k$ was termed the real action field.
-> were
The subject of the sentence is "The pixel-wise differences" (plural). Therefore, the verb should be were (plural), not "was" (singular).

Line 727-732: For $i ∈ N$ and $S ⊆ N \ {i}$, the interaction effect of the pattern $S ∪ {i}$ is equal to the interaction effect of $S$ with the presence of $i$ minus the interaction effect of $S$ with the absence of $i$, i.e., $∀S ⊆ N \ {i}, I(S ∪ {i}) = I(S|iis always present) - I(S).I(S|iis always present)$ denotes the interaction effect when the variable i is always present as a constant context, i.e. $I(S|iis always present) = ∑ S⊆I(S) (-1)|S| · u(L ∪ {i})$.
-> … $= I(S|i is always present) - I(S). I(S|i is always present)$ …
The period (.') immediately following $I(S)$ unnaturally breaks the sentence, causing the end of the equation to be jumbled with the start of the explanation.
It should be ... $- I(S). I(S|i is always present)$ denotes..., with a space inserted after the period to properly begin a new sentence.

**Questions:**

I have mentioned the concerns in the Weakness section.

---

> ### Author Response · Authors · 2025-12-02
> **Rebuttal to Reviewer 7rhP**
>
> ## Response
>
> $\colorbox{#FFF9C4}{Q1: “The current experimental results can only be considered valid for a biased subset of the model's output (i.e., the foreground)”.}$
>
> Thank you. We have followed your suggestions to conduct **a new experiment** on patches randomly selected from the entire image, including background regions. Specifically, given an image $x$ generated by the BigGAN, we selected an arbitrary patch set $N’$, which is different from the foreground-focused patch set $N$ discussed in the paper. We then use the following two metrics to evaluate the faithfulness of the representation structure corresponding to $N’$ by analyzing the specific regional patterns associated with these newly selected patches:
>
> - To verify whether the extracted regional patterns $\Delta x_k$ are spatially located in its action fields $A_k$, we used the metric $\gamma= \Vert region(A_k \mid \Delta x_k)\Vert_2/\Vert \Delta x_k \Vert_2 $ (proposed in Paragraph 1 of Section 3.2.1, Line 362) to validate the spatial boundedness property. The metric $\gamma$ evaluates the strength of the regional pattern Δx_k within the action field A_k relative to its total strength ||Δx_k||. A higher $\gamma$ means the regional pattern Δx_k is predominantly contained within $\Delta f_k$’s action field A_k，indicating the regional patterns satisfy the spatial boundedness property.
> - To verify whether the extracted regional patterns $\Delta x_k$ is consistent under different surrounding contexts $S$, we used the metric $\alpha= \Vert \Delta x_{k,S} - \Delta x_{mean} \Vert_2/\Vert \Delta x_{mean}\Vert_2 $ （proposed in Paragraph 1 of Section 3.3, Line 417）to validate the consistency property. The metric $\alpha$ evaluates the relative deviation of the regional patterns $\{\Delta x_{k,S}\}$ generated by $\Delta f_k$ conditioned on varying patch sets $S$ outside the action field $A_k$, i.e., $S \cap A_k= \emptyset$. A lower $\alpha$ value indicates that the regional patterns $\{\Delta x_{k,S}\}$ under different contexts $S$ remain consistently close to the mean regional pattern $\Delta x_{k,mean}$, demonstrating that the regional patterns satisfy the consistency property.
>
> Experimental results based on the newly selected patches $N’$ are shown in Figure 17 of Appendix K of the revised paper:
>
> - Spatial Boundedness: All regional patterns $\Delta x_k$ exhibited $\gamma$ values greater than 0.98, indicating they were predominantly contained within their respective action fields $A_k$ (Fig. 17(a)). The results demonstrated that the representation structure corresponding to the newly selected patches in $N’$ maintained strong spatial boundedness.
>
> - Consistency: The $\alpha$ values of the regional patterns $\Delta x_k$ were smaller than 0.09. The low $\alpha$ values confirmed that the feature component $\Delta f_k$ consistently generated almost same regional patterns $\{\Delta x_{k,S}\}$ even when the surrounding context $S$ changed (Fig. 17(b)). Results validated that the representation structure corresponding to $N’$ maintained strong consistency.
>
> These results confirmed that the key properties (spatial boundedness and consistency) held for arbitrary sets of patches, demonstrating that our method was generally applicable and not limited to a biased subset (e.g., foreground) of the generated image.
>
> $\colorbox{#FFF9C4}{Q2: Concern about the asymmetric comparison ($l_a \neq l_b$) when evaluating the transferability.}$
>
> Thank you. Following your suggestion, we have redesigned the metric to ensure a symmetric comparison for transferability, as follows:
>
> Specifically, given a pair of generated images $(\mathbf{x}^{(a)}, \mathbf{x}^{(b)})$ containing the same object, we selected the same number (top-$l$) of feature components with the highest L2-norm from both images. Let $D_{\mathbf{x}^{(a)}}$ and $D_{\mathbf{x}^{(b)}}$ denote the sets of the corresponding action fields, respectively.
>
> We then quantified transferability using the transferability score $t = \frac{|D_{\mathbf{x}^{(a)}} \cap D_{\mathbf{x}^{(b)}}|}{l}$. This metric measures the proportion of feature components extracted from $\mathbf{x}^{(a)}$ that could be transferred to the representation of image $\mathbf{x}^{(b)}$. By fixing the number of selected components ($l$) for both sides, this new metric guarantees a fair and symmetric assessment.
>
> Experiments on NVAE-generated image pairs, as detailed in Figure 18 of Appendix L, showed that under this symmetric metric, the transferability score $t$ was 0.85. This high $t$ value confirmed that a significant majority of salient regional patterns were indeed effectively transferred across images of the same object.
>
> $\colorbox{#FFF9C4}{Q3: Concern about two grammar mistakes: “Line 358：The verb should be were (plural), not "was" (singular).”,}$
> $\colorbox{#FFF9C4}{ “Line 727-732：Missing a space inserted after the period to properly begin a new sentence”}$
>
> Thank you. We have corrected these grammar mistakes in the revised manuscript.

---

### Official Review · Reviewer_qwp8 · 2025-10-31

**Soundness:** 2
**Presentation:** 3
**Contribution:** 2
**Rating:** 4
**Confidence:** 3

**Summary:**

The paper presents three axioms; feature completeness, spatial boundedness and consistency, to explain the internal representation of an image. An intermediate feature f can be decomposed as a base f0​ plus a sum of feature components {Δfk}. Each component has an action field Ak​ (the image patches it affects) and contributes a regional pattern Δx​, thus, the image is explained as a superposition of these regional patterns.

**Strengths:**

- The work formalizes a representation structure via three axioms: feature completeness, spatial boundedness, and consistency. The feature decomposition turns an intermediate feature into a set of localized components with precise action fields.

- The paper provides interesting insights into the intrinsic representation. The analysis indicates high sparsity of the extracted feature representation, meaning that a region can be explained by a small number of feature components while the contribution of the others is negligible.

**Weaknesses:**

- Human interpretability gap: while the paper “disentangles” the representation into minimal regional features, this does not automatically yield human-interpretable factors. The extracted components remain complex. Minimal for reconstruction ≠ meaningful for humans. A stronger claim of interpretability would require concept aligned minimality, e.g. components that are minimal for foreground vs. background, hair vs. skin, sky vs. building, etc. Extending the framework to be minimal with respect to human-interpretable concepts would better support both interpretability and controllability of generation.
- Missing computational analysis: compute resources are listed, analysis isn’t. Beyond acknowledging cost and describing setup choices, the paper does not quantify end-to-end time per image/model. The high computational cost makes practical applicability questionable.
- Overlap in action fields: action fields heavily overlap, and the feature components are not orthogonal,making attribution within shared regions ambiguous. The paper offers no weighting/attribution rule or orthogonality control to resolve this.
- Minor weakness; evaluation on diffusion models is limited: The diffusion evaluation is restricted to a single-step model on small resolution images (64x64).

**Questions:**

See weaknesses

---

> ### Author Response · Authors · 2025-11-22
> **Rebuttal By Authors**
>
> ## Response
>
> Thank you. Please find our responses to your questions below. For questions that remain unanswered, we will respond in a few hours.
>
> $\colorbox{#FFF9C4}{Q1: How can the framework be extended to be minimal with respect to human-interpretable concepts?}$
>
> A good question. In the field of explainable AI, (1) explaining towards human cognition and (2) objectively explaining the truth of internal mechanisms have always been two different directions and schools of thought. **Your question is essentially a conflict between two schools of thought. Explaining objective mechanisms often implies that the resulting explanation does not fully align with human cognition.** For example, mechanistic interpretability studies [1, 2, 3] attempt to explain the meaning of specific neurons, but there has never been a theory to strictly guarantee that inference patterns autonomously encoded by these neurons fully match human cognition.
>
> Therefore, in the field of interpretability, ensuring the objectivity of the explanation will inevitably and partially hurt its alignment with human cognition, to some extent. The situation is similar in our paper. We aim to explain the representation structure objectively modeled by the neural network, but to a certain extent, this makes it difficult to ensure that the regional patterns extracted by our method fully align with human cognition.
>
> Nevertheless, although we did not intentionally require the extracted feature components to model semantically related patterns in our algorithm design, we found that the patterns extracted using our method are partially consistent with human cognition. Please see Figures 1,2, and 8 in the paper for the visualization of regional patterns extracted using our method.
>
> Furthermore, lots of experiments have indicated that the internal representation structure of the neural network extracted by our method is faithful.
> - The experiments in Section 2.3 verify the sparsity of feature components, i.e., the feature components tend to be sparse. Please see Figure 3 for details.
> - The experiments in Section 3.4 verify the transferability of regional patterns, i.e., the regional patterns triggered on an image are highly transferable to another image. Please see Figure 8 for details.
>
> [1]Bricken, T., et al. (2023). Towards monosemanticity: Decomposing language models with dictionary learning. Transformer Circuits Thread, Anthropic. [https://transformer-circuits.pub/2023/monosemantic-features/index.html](https://transformer-circuits.pub/2023/monosemantic-features/index.html)
>
> [2]Anthropic. (2024). Mapping the mind of a large language model. Anthropic Research. [https://www.anthropic.com/research/mapping-mind-language-model](https://www.anthropic.com/research/mapping-mind-language-model)
> [3] Language Model Interpretability team. (2024, July 31). Gemma Scope: Helping the safety community shed light on the inner workings of language models. Google DeepMind. [https://deepmind.google/blog/gemma-scope-helping-the-safety-community-shed-light-on-the-inner-workings-of-language-models/](https://deepmind.google/blog/gemma-scope-helping-the-safety-community-shed-light-on-the-inner-workings-of-language-models/)
>
> $\colorbox{#FFF9C4}{Q2: How to address “ambiguous attribution from shared regions”, since “action fields heavily overlap, and the feature components are not orthogonal”?}$
>
> A good question. Spatially overlapped features are not necessarily equivalent to non-orthogonal features. Although the action fields of these feature components are spatially overlapped, the regional patterns of these feature components $\{\Delta f_k\}$ are well-disentangled. That is, for $\Delta f_1$ and $\Delta f_2$, although their corresponding action fields $A_1$ and $A_2$ are partially overlapped, the vector direction of $\Delta f_1$ is approximately orthogonal to the vector direction of $\Delta f_2$.
>
> Specifically, the image change $\Delta x_k$ caused by a feature component $\Delta f_k$ is independent of other components $\Delta f_j$ (j ≠ k), according to Axiom 3: the consistency axiom (proposed in Section 2.2, Line 181 and experimentally verified in Section 3.3). Each feature component $\Delta f_k$ is supposed to consistently add the same regional pattern $\Delta x_k$ to the generated image $x$, no matter how other feature components $\Delta f_j$ (j ≠ k) are set. That is, $\forall \Omega \subseteq M\setminus\{k\},\quad \mathcal{G}(\Omega \cup \{k\})-\mathcal{G}(\Omega) = \Delta x_k$, where $M$ denotes the set of all feature components, $\mathcal{G}(\Omega)$ denotes the image generated by adding all feature components in $\Omega$ to the intermediate layer. The generated image $x$ is equal to summing up all the regional patterns, i.e., $x=x_0 + \sum_{k \in M}\Delta x_k$ , where $x_0$ denotes the base image generated by the base feature $f_0$.

---

> > ### Comment · Reviewer_qwp8 · 2025-11-27
> >
> > I thank the authors for clarifying the intended scope of the paper, in particular that the goal is a mechanistic explanation of the internal representation structure rather than full alignment with human-level concepts. This helps me better understand how the work is positioned.
> >
> > However, my main concern about practical applicability remains. The method appears computationally demanding, and from the current experiments it is still unclear whether it can scale beyond the relatively small setups considered in the paper. Moreover, the need to restrict the analysis to a small subset of patches due to computational cost makes it difficult to assess how general the claimed representation structure really is.
> > For these reasons, I will maintain my original score.

---

> ### Author Response · Authors · 2025-12-02
> **Rebuttal to Reviewer qwp8**
>
> Thank you for your feedback. To address your remaining concerns regarding computational complexity and how general the proposed representation structure is, we have conducted a **detailed computational complexity analysis** and performed **additional experiments** to demonstrate that our method is feasible for full-image analysis through a grouping strategy.
>
> **Detailed responses**:
>
> $\colorbox{#FFF9C4}{Q3: “Restricting the analysis to a subset of patches makes it difficult to assess how general the representation structure is.” “Compute analysis isn’t listed”.}$
>
> 1. **Computational Complexity and Acceleration Strategies**: The core step of our algorithm involves the optimization of the minimal feature $\hat{f}_S$ for each subset $S$ of the target patch set $N$. Although the theoretical time complexity for this step is $O(2^{|N|})$, we can apply a strategy to effectively prevent the exponential increase in computational load.
>
>     - **Method**: We partition the image into $K$ overlapping local groups of size $n'$ (where $n' \ll |N|$). This reduces the complexity from exponential $O(2^{|N|})$ to linear-scale $O(K \cdot 2^{n'})$.
>
>     - **Empirical Validation**: As detailed in Appendix J, with $n'=9$ and multi-process parallelization, we can compute the structure for a full image in ~30 minutes. This proves our method can scale beyond small patch subsets to full-image analysis.
>
> 2. **Engineering Implementation vs. Theoretical Exploration**: In the field of explainable AI (XAI), there are broadly two schools of thought:
>     - One focuses on engineering-driven implementation of specific explanation demands, such as CAM-based methods (e.g., Grad-CAM, Score-CAM), which are computationally efficient but often lack formal guarantees regarding the faithfulness of their explanations.
>     - The other school explores the rigorously formulating and mathematically proving the existence of an ideal explanation. An example is the Shapley value, which defined four axioms (i.e., Efficiency, Symmetry, Dummy, and Additivity) to theoretically ground the definition of ideal attribution values，despite its inherent high computation cost of O(2^n).
>
>     Similarly with the latter, we aim to define the faithful representation structure for image generation through a set of axiomatic properties. While this theoretical contribution comes at the cost of complexity, it is a reasonable price for foundational progress. Just as the computational cost of Shapley values does not diminish their value, The computational complexity of our method is necessary for rigorously defining the image generation representation structure. Future engineering-focused research can build upon this foundation to develop approximation methods.
>
> $\colorbox{#FFF9C4}{Q4. Minor: “The diffusion evaluation is restricted to a single-step model on small resolution images (64x64).”}$
>
> Thank you. Our method is primarily applicable to single-pass, feed-forward generative models (e.g., GANs, VAEs, one-step diffusion models). Applying our optimization-based method directly to standard, multi-step diffusion models introduces significant computational complexity. Backpropagating gradients through the long chain of iterative denoising steps drastically increases time complexity per feature optimization step and leads to high memory consumption for storing the computation graph, especially at high resolutions.
>
> Therefore, extending our framework to efficiently compute the representation structure for iterative models poses significant challenges, and we plan to address it in future work.

---

### Official Review · Reviewer_VxFz · 2025-11-01

**Soundness:** 3
**Presentation:** 3
**Contribution:** 4
**Rating:** 6
**Confidence:** 3

**Summary:**

This paper addresses the problem of interpretability in generative DNNs, arguing that most current methods are "superficial"  because they only analyze the meaning of isolated feature vectors rather than a complete "representation structure". To address this, the authors first propose a rigorous theoretical framework, defining a set of three axiomatic properties that any faithful explanation of a generative model's representation structure must satisfy.
The paper's core technical contribution is to prove that feature components satisfying these axioms can be formally computed. The method is validated on four generative models (SiD Diffusion, NVAE, BigGAN, and StyleGAN). The experiments are specifically designed to verify that the extracted components successfully adhere to the proposed axioms (Linear Additivity , Spatial Boundedness , and Consistency ).

**Strengths:**

The paper's most significant contribution is conceptual. It is not just another explanation method but a formal definition of what a faithful, complete explanation of a generative model's internal structure is. By establishing the three axioms (Completeness, Boundedness, Consistency) , the authors provide a rigorous mathematical foundation for a field that has largely relied on "superficial"  and incomplete analyses. The experimental design is excellent and serves as a model for theory-driven research. Instead of just showing qualitative results, the authors systematically test each of their theoretical claims.

**Weaknesses:**

Similar to many interpretability studies, the current analysis provides a useful “window” into understanding the generative model. However, the explanation relies on a relatively narrow theoretical perspective, which may limit its generalizability. For most researchers in generative modeling, there remains a conceptual gap: how can the generative process be understood at a higher, more intuitive level? Developing a simpler, working model that builds on the current interpretations could help a broader audience grasp the key mechanisms.

**Questions:**

1. Regarding the definition and selection of features, to what extent do the chosen priors constrain the completeness of the model interpretation? Could alternative feature choices reveal additional insights?
2. For researchers aiming to build better generative models, the practical value of the framework is also important. Beyond explaining a trained model, it would be valuable to discuss how these disentangled primitives could be leveraged to improve generative models—for example, enabling finer-grained controllable editing, diagnosing biases, or enhancing generalization.

---

> ### Author Response · Authors · 2025-12-04
> **Rebuttal to Reviewer VxFz**
>
> ## Response
>
> $\colorbox{#FFF9C4}{Q1: How does “the chosen priors constrain the completeness of the model interpretation?”}$
>
> Thank you. In Sections 3.1–3.3, **we have experimentally demonstrated** that the completeness of the model interpretation holds across various mainstream image generation architectures (e.g., VAE, GAN, and Diffusion Models) trained on different datasets. Specifically, the representation structures extracted from these diverse architectures all satisfy the three axiomatic properties proposed in the paper (feature completeness, spatial boundedness and consistency). Please refer to the experimental results in Figure 5,6,7.
>
> $\colorbox{#FFF9C4}{W: “The explanation relies on a relatively narrow theoretical perspective, which may limit its generalizability.”}$
> $\colorbox{#FFF9C4}{Q2: “How these disentangled primitives could be leveraged to improve generative models? ”}$
>
> **1. Theoretical Explainability vs. Engineering Heuristics.** We respectfully clarify that mechanistic explanation and pragmatic engineering techniques represent different research directions in explainable AI. From the perspective of fundamental research, objectively explaining the representation structure of image generation networks remains a significant challenge and a theoretical void.
>
> While previous works have examined the internal mechanisms of classification models (e.g., using Shapley values) or visualized features in intermediate layers (e.g., OpenAI’s Microscope [1], Google’s Deep Dreaming with BERT [2]), our work is the first to extract a concise and globally interpretable representation structure that encompasses all feature components to precisely explain the image generation networks. Crucially, we have experimentally discovered two fundamental properties: the sparsity of feature components and the transferability of regional patterns. These findings suggest that the internal representation structure extracted by our method is faithful.
>
> **2. Potential Applications**: We have **conducted a preliminary experiment** into model diagnostics and debugging, regarding how these primitives can improve generative models. Please see Appendix I for details.
>
> As detailed in Appendix I, we compared the internal representation structures of well-trained networks against those of under-trained networks. We observed that well-trained networks exhibit a sparser representation structure. Consequently, these disentangled primitives can serve as a metric for debugging the training process of image generation networks.
>
> Reference:
>
> [1] OpenAI. (2020, April 14). Microscope. https://openai.com/index/microscope/
>
> [2] Bäuerle, A., & Wexler, J. (n.d.). Deep dreaming with BERT. People + AI Research (PAIR), Google. https://pair-code.github.io/interpretability/text-dream/blogpost/

---

### Author Response · Authors · 2025-12-04
**Summary for Area Chair**

We sincerely thank the reviewers for their time and constructive feedback. A summary of the key concerns raised by the reviewers and our corresponding responses is provided below:

**(1) Question**: Can the faithfulness of the representation structure be evaluated based on its performance in engineering tasks (e.g., image editing)?

**Our Answer: No**. Our work does not rely on engineering heuristics for interpretation; rather, it is grounded in a rigorous theoretical framework. We have introduced three axiomatic properties in the paper to formally evaluate and guarantee the faithfulness of the representation structure.

**(2) Question**: Is the proposed method limited to interpreting only specific sets of image patches?

**Our Answer: No**. We conducted **additional experiments** to verify that our method is capable of interpreting arbitrary sets of image patches.

**(3) Question**: Do the extracted patterns align with human perception?

**Our Answer: Yes**. As demonstrated by the visualization results in Figure 1,2, and 8 in the paper, the extracted patterns exhibit semantic shapes (e.g., eyes, nose, and mouth) that are consistent with human perception.

---

### Meta-Review · Area_Chair_eWZh · 2025-12-22

**Summary:**

The submission aims to disentangle primitive feature components from intermediate-layer features for image generation.  Reviewers raised questions about the submission's motivation, practical applicability and generalizability, and presentation.

**Reviewer Concerns:**

Reviewers felt clearer about the motivation post rebuttal, but their concerns regarding the practical applicability and generalizability remain and seem not addressable given the scope of the submission.

**Reviewer Scores:**

Reviewers will most likely maintain their scores of 6, 4, 4, 4.

---

### Decision · Program_Chairs · 2026-01-26

Reject